# Inhibition of overactive TGF-β attenuates progression of heterotopic ossification in mice

Xiao Wang [1], Fengfeng Li[2], Liang Xie[1], Janet Crane[1], Gehua Zhen[1], Yuji Mishina [3], Ruoxian Deng[1], Bo Gao[1], Hao Chen[1], Shen Liu[1,2], Ping Yang[1], Manman Gao[1], Manli Tu[1], Yiguo Wang[1], Mei Wan[1], Cunyi Fan[2] & Xu Cao[1]

Acquired heterotopic ossification (HO) is a painful and debilitating disease characterized by extraskeletal bone formation after injury. The exact pathogenesis of HO remains unknown. Here we show that TGF-β initiates and promotes HO in mice. We find that calcified cartilage and newly formed bone resorb osteoclasts after onset of HO, which leads to high levels of active TGF-β that recruit mesenchymal stromal/progenitor cells (MSPCs) in the HO microenvironment. Transgenic expression of active TGF-β in tendon induces spontaneous HO, whereas systemic injection of a TGF-β neutralizing antibody attenuates ectopic bone formation in traumatic and BMP-induced mouse HO models, and in a fibrodysplasia ossificans progressive mouse model. Moreover, inducible knockout of the TGF-β type II receptor in MSPCs inhibits HO progression in HO mouse models. Our study points toward elevated levels of active TGF-β as inducers and promoters of ectopic bone formation, and suggest that TGF-β might be a therapeutic target in HO.

---

[1] Department of Orthopedic Surgery, School of Medicine, Johns Hopkins University, Baltimore, MD 21205, USA. [2] Department of Orthopedic Surgery Shanghai Sixth People's Hospital 200030 Shanghai, China. [3] School of Dentistry, University of Michigan, Ann Arbor, MI 48109 USA. Correspondence and requests for materials should be addressed to X.C. (email: xcao11@jhmi.edu)

Heterotopic ossification (HO) is an ectopic formation of the bone in extraskeletal tissues. It severely incapacitates people in their daily life[1]. HO is mostly acquired, but in rare instances they may be congenital. Acquired HO develops as a common clinical complication after trauma, including fractures, total hip arthroplasty, deep burns, and central nerve system injury, and results in a high prevalence rate[2]. The patho-mechanism of acquired HO is unknown[1,3]. Therefore, clinical therapy is limited to radiation and/or surgical excision for already formed HO, which is associated with an extremely high recur-rence rate (radiologically 82–100%, clinically 17–58%) and frequent complications[1–3]. HO is also seen in rare genetic diseases such as fibrodysplasia ossificans progressive (FOP)[4] and progressive osseous heteroplasia (POH)[5]. The genetic cause of FOP has been identified as a heterozygous R206H mutation in the bone morphogenetic protein (BMP) type I receptor, activin receptor-like kinase 2 (ALK2) in classic FOP patients (98%)[6]. There is no known treatment for FOP in clinical practice[7]. POH is a process of intramembranous bone formation by heterozygous mutations in *GNAS* in patients[8].

Histologically, acquired HO and FOP are believed to develop through a process of endochondral ossification involving four stages: inflammation, chondrogenesis, osteogenesis, and matura-tion[1]. A variety of cells participate in HO including tissue-resident mesenchymal, vascular, circulating, hematopoietic, and bone marrow-derived cells[9–11] regulated by intricate signaling pathways. In the inflammation stage of HO, immune cells infil-trate the site[12]. In an FOP mouse model, expression of con-stitutively active *ALK2* in endothelial cells causes endothelial-to-mesenchymal transition (EndMT) and acquisition of a stem cell-like phenotype[10]. In the chondrogenesis stage, the mesenchymal stem/progenitor cells (MSPCs) differentiate into chondrocytes[13], further confirmed by a recent lineage tracing study in FOP mouse models that demonstrated that $Scx^+$ tendon-derived progenitors and muscle-resident interstitial $Mx1^+$ cells give rise to chondrocytes in HO lesions in the chondrogenesis stage[9]. In the osteogenesis stage, chondrocytes undergo hypertrophy and calcification, followed by invasion of blood vessels for ectopic bone formation[14,15]. In the final maturation stage, fully developed cancellous bone with marrow is formed.

Transforming growth factor beta (TGF-β) subfamily only has three closely related isoforms, TGF-β1, β2, and β3. TGF-βs are expressed with the latency-associated protein (LAP), rendering it inactive by masking the extracellular matrix (ECM) in many different tissues[16]. TGF-βs are only present in mammals and are important for tissue remodeling and/or repair to maintain tissue homeostasis[17–19]. Many diseases in different organs are asso-ciated with aberrant activation or elevated levels of TGF-β, such as fibrosis of skin, kidney, lung, liver, and metastasis of different tumors[20]. In the skeleton, active TGF-β is released during osteoclastic bone resorption to recruit stem cells to couple bone resorption for bone remodeling[17]. Loss of the spatial and tem-poral TGF-β signaling results in several complications, including Camurati–Engelmann disease (CED)[17,21], Loeys–Dietz syn-drome[22], Shprintzen–Goldberg syndrome[23], Marfan syndrome[24], osteogenesis imperfecta[25], and osteoarthritis[18]. Osteogenesis is a metabolically demanding process supported by angiogenesis[26]. Abundant blood vessels are also seen during the progression of acquired HO[1].

We have previously demonstrated that PDGF-BB secreted by tartrate-resistant acid phosphatase-positive (TRAP+) pre-osteoclasts recruits endothelial progenitors and MSPCs to couple CD31$^{high}$Emcn$^{high}$ blood vessels (termed type H vessels) with osteogenesis[26–28]. In this study, we found high levels of active TGF-β increases MSPC number and drives progression of HO, including acquired HO and FOP. PDGF-BB concentration were also increased during HO progression. Inhibition of TGF-β activity effectively attenuated HO progression in different rodent models including FOP.

## Results

**TGF-β activity is elevated in human HO patients.** To determine the pathogenesis of HO, we examined surgical specimens of acquired HO, identified radiographically from patients after internal fixation for elbow fracture or after central nervous system trauma (CNST) at osteogenesis stage (3–4 months after initial injury) and maturation stage (14–16 months after initial injury). H&E and Safranin O and fast green (SOFG) staining of the HO specimens revealed thick cartilage layers adjacent to cancellous bone and marrow at the osteogenesis stage. By contrast, at maturation stage, larger well-developed cancellous bone and marrow with thinner proteoglycan-enriched cartilage layers were observed (Fig. 1a, b). TRAP staining revealed that the number of TRAP+ preosteoclasts and osteoclasts was significantly higher at osteogenesis stage and decreased at maturation stage (Fig. 1c, d). The number of phosphorylated Smad2/3-positive (pSmad2/3+) cells, a TGF-β downstream signaling transducer, was significantly elevated at osteogenesis stage and lowered at maturation stage (Fig. 1e, f). Additionally, immunostaining demonstrated that the number of PDGF-BB+ cells were significantly increased at the osteogenesis stage of HO and decreased at maturation stage (Fig. 1g, h). Significantly elevated levels of active TGF-β1 and PDGF-BB in serum were seen in the osteogenesis stage HO patients relative to the maturation stage, both of which were a higher concentration than healthy controls (Fig. 1i, j). Immu-nostaining of human Mesenchymal stem cell (MSC) markers CD73 and CD90 revealed that the number of MSCs in the HO bone marrow was significantly increased at the osteogenesis stage and decreased at the maturation stage (Fig. 1k, l). Altogether, our results reveal that acquired HO in human progresses via endo-chondral ossification, with high levels of active TGF-β, PDGF-BB, and MSCs in the HO microenvironment.

**Active bone remodeling during HO progression in mouse HO model.** To examine the mechanism of HO osteogenesis, we uti-lized a trauma-induced HO mouse model by percutaneous Achilles tendon puncture (ATP)[29]. Heterotopic bone was formed in the trauma area of the mouse model 6 s post puncture and continued to enlarge up to 15 weeks (Fig. 2a, Supplementary Fig. 1a). H&E, SOFG staining, and immunostaining of type II collagen (COLII) showed typical endochondral ossification after ATP (Fig. 2b, Supplementary Fig. 1b, c). Abundant proteoglycan and COLII were observed 3 weeks post puncture and gradually moved from the HO center to the outer layer of the newly formed bone adjacent to the soft tissues (Fig. 2b, Supplementary Fig. 1b, c). Immature woven bone was present at 6 weeks and a fully developed cancellous bone with marrow was noted from 9 to 15 weeks (Fig. 2b, Supplementary Fig. 1b, c). Similar to human HO, TRAP staining showed that the number of TRAP+ cells was increased in the heterotopic bone 6 weeks post puncture, and the continued osteoclastic bone resorption generated large bone marrow cavities by 15 weeks when few TRAP+ cells were seen (Fig. 2c, d). The number of pSmad2/3+ cells increased as early as 3 weeks post puncture and were maintained at a high con-centration before a reduction by 15 weeks (Fig. 2e, f).

The number of PDGF-BB+ cells was also high during HO progression from 3 weeks after ATP, plateauing at 6 weeks followed by a decrease at 15 weeks (Fig. 2g, h). The concentra-tions of active TGF-β1 and PDGF-BB in serum were elevated from 3 weeks after surgery, peaked at 9 weeks, and returned to baseline levels (Fig. 2i, j). Nestin+ cells are a subgroup of MSPCs

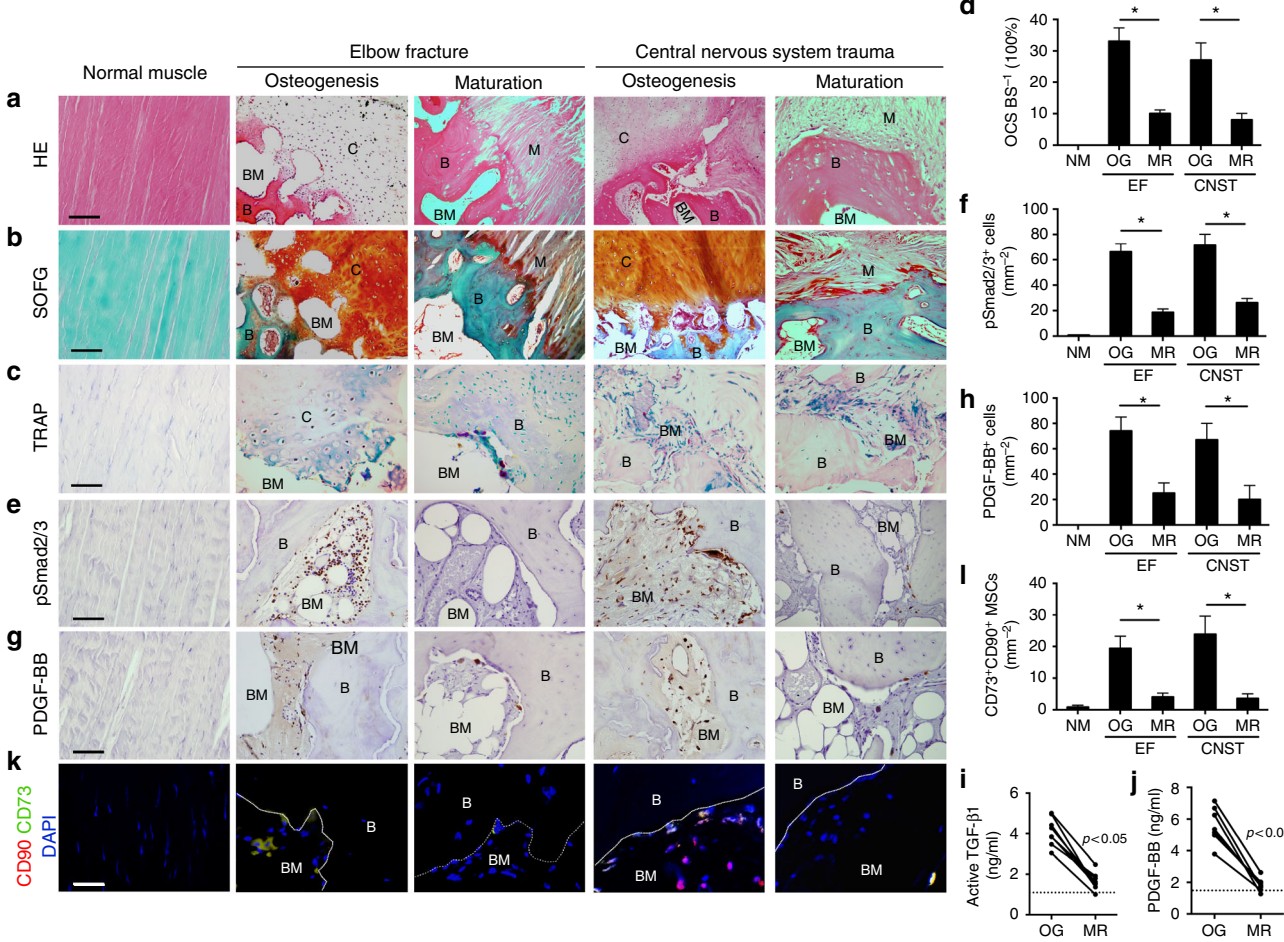

**Fig. 1** Elevated active TGF-β levels in human HO. **a** H&E staining of normal muscle (left) and HO in early osteogenesis stage (middle) and late osteogenesis stage (right) after elbow fracture (EF) and central nervous system trauma (CNST). Scale bar, 100 μm. **b** Safranin O and Fast Green (SOFG) staining of normal muscle (left) and HO. Proteoglycan (red) and heterotopic bone (green). Scale bar, 100 μm. **c** TRAP⁺ cells (red) and **d** quantitative analysis of the number of TRAP⁺ osteoclast surface (OCS) per bone surface (BS). Scale bar, 100 μm. Immunohistochemical staining of **e** pSmad2/3⁺ cells (brown) and **g** PDGF-BB⁺ cells (brown) and quantitative analysis of the number of **f** pSmad2/3⁺ cells or **h** PDGF-BB⁺ cells per bone marrow area (mm²). **i**, **j** Quantitative analysis of **i** active TGF-β and **j** PDGF-BB in serum in HO patients at early osteogenesis stage and late mature stage by ELISA. Dotted line indicates the average concentration of healthy people with active TGF-β of 1.23 ng/ml and PDGF-BB of 1.68 ng/ml. **k** Immunofluorescent staining and **l** quantitative analysis of CD73⁺ (green) and CD90⁺ cells (red) in bone marrow of ectopic bone marrow in HO patients. Blue color indicates DAPI staining of nuclei. Scale bar, 20 μm. C cartilage, B bone, BM bone marrow, M muscle, NM normal muscle, EF elbow fracture, CNST central nervous system trauma, OR osteogenesis MR maturation. All data are shown as the mean ± s.d. n = 9 per group for **d**, **f**, **h**, **l** histomorphometry analysis. *p < 0.05 as determined by unpaired, two-tailed Student's t-test. n = 8 per group for **i**, **j** ELISA analysis. *p < 0.05 as determined by paired, two-tailed Student's t-test

in adult marrow and Nestin is also expressed in proliferating endothelial progenitor cells[30–33]. Nestin⁺ blood vessels in bone marrow are reported to be associated with calcified bone[32]. Immunostaining for Nestin revealed a significantly higher number of Nestin⁺ cells in the HO bone marrow of ATP-induced HO mice compared to sham controls (Fig. 2k, l). Immunostaining for CD31 and Emcn revealed that osteogenic CD31^high Emcn^high (type H) vessels surrounded the cartilage formed in Achilles tendon 3 weeks after ATP, while Emcn⁺CD31⁻ vessels were located at the outer layer (Fig. 2m), but not in the cartilage. Type H vessels were identified in the HO bone marrow 6 weeks after ATP and significantly higher in ATP-induced HO mice relative to sham controls (Fig. 2m, n) indicating an osteogenesis stage as type H vessel formation is specifically coupled with new bone formation[26]. The number of type H vessels decreased to approximately baseline levels of sham-operated mice by week 15 (Fig. 2m, n). Osteocalcin positive (Ocn⁺) osteoblasts were present 6 weeks post puncture for de novo bone formation and decreased by 15 weeks (Fig. 2o, p).

Taken together, the ATP-induced HO mouse model manifests a similar mechanism as observed in the human HO specimens, implying that high concentrations of active TGF-β may contribute to the pathogenesis of HO.

**Elevated TGF-β at the inflammation stage triggers HO.** Inflammation is the initiation stage of HO development. We found that accumulation of CD68⁺ immune cells and high levels of active TGF-β four days after HO induction in ATP mice (Fig. 3a–d), suggesting that active TGF-β and immune cells are closely related to the onset of HO. It has been reported that inflammatory stimuli enhance the release of active TGF-β by macrophages[34]. To elaborate the role of macrophages in initiation of HO, we first analyzed HO formation in colony-stimulating factor-1 (CSF-1)-deficient (Csf1⁻/⁻) mice, which have macrophage deficiency, as CSF-1 is essential for the survival of mono-cyte-macrophage–lineage cells. HO was inhibited 6 weeks after HO induction by ATP (Fig. 3e, f). We then generated a LysM-cre:: Tgfb1^flox/flox mouse model (Tgfb1⁻/⁻), where immune cells,

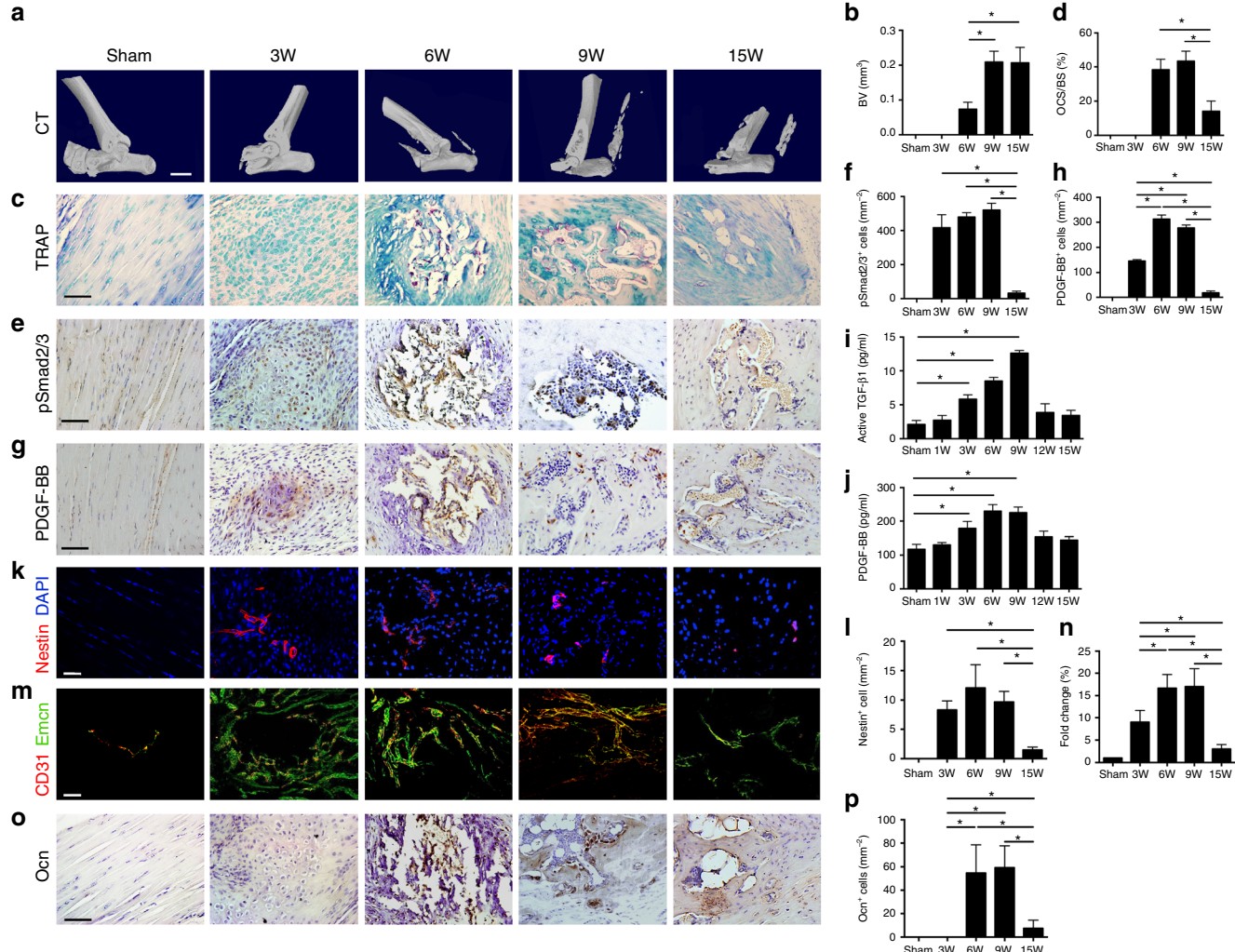

**Fig. 2** Elevated active TGF-β levels were associated with increased angiogenesis in HO mice. **a** Micro CT images of Achilles tendon (sagittal view) after sham operation or at 3, 6, 9, and 15 weeks after ATP and **b** quantitative analysis. Scale bar, 2 mm. **c** TRAP staining (magenta) and **d** quantification of heterotopic bone in mouse Achilles tendon. **e**, **g** Immunohistochemical staining and **f**, **h** quantification of **e**, **f** pSmad2/3[+] cells and **g**, **h** PDGF-BB[+] cells after sham operation or ATP. Scale bar, 50 μm. **i**, **j** Quantitative analysis of **i** active TGF-β and **j** PDGF-BB in serum determined by ELISA. **k** Nestin[+] (red) cells in the ectopic bone marrow of sham or ATP mice and **l** quantification. Scale bar, 50 μm. Blue indicates DAPI staining of nuclei. **m** Emcn[+] (green) and CD31[+] (red) cells and **n** quantification of the fold change of type H vessels in ATP mice normalized to that of sham mice (set to 1) in the ectopic bone marrow. Scale bar, 100 μm. Yellow indicates type H vessels. **o** Immunohistochemical staining and **p** quantification of Ocn[+] cells after sham operation or ATP. All data are shown as the mean ± s.d. n = 8 per group. *p < 0.05 as determined by one-way ANOVA

including macrophage/monocyte lineage and neutrophils[35,36] no longer produce TGF-β1. Again, 6 weeks after HO induction by ATP, no HO was formed in *Tgfb1*[−/−] mice, while evident of HO formation in *Tgfb1*[flox/flox] control mice (Fig. 3g, h). Therefore, it is the elevated TGF-β secreted by immune cells at the inflammation stage that triggers HO.

To further validate the role of elevated active TGF-β in HO, we examined TGF-β transgenic mice with a CED-derived TGFB1 mutation (H222D), where expression of high active TGF-β concentrations is driven by type I collagen (COLI) promoter (termed CED mice)[18]. HO formed spontaneously in Achilles tendons, in which COLI is abundantly found, in 4-month-old CED mice (18 of 22 mice), while no ectopic bone was noted in wild-type (WT) littermates (Fig. 3i–k) (0 out of 22 mice). Ossifications were also detected in ligaments of paws, likely because Achilles tendons and ligaments of paws have abundant COLI expression with higher active TGF-β levels and are actively used with bearing more mechanical loading prone to pressure or injury (Supplementary Fig. 2). A significantly higher number of

Nestin[+] cells was observed in the HO bone marrow of CED mice relative to their WT littermates (Fig. 3l, m). As expected, the number of osteogenic CD31[high]Emcn[high] type H vessels in HO was significantly higher in CED mice relative to their WT littermates (Fig. 2n, o). In addition, Ocn[+] osteoblast number was significantly higher in the HO bone marrow of CED mice relative to WT littermates (Fig. 3p, q). Altogether, CED mice have an HO phenotype in Achilles tendon similar to ATP mouse models, indicating that high levels of active TGF-β is the driving force for the pathogenesis of HO.

**TGF-β antibody attenuates progression of ATP-induced HO.** We next examined whether inhibition of TGF-β activity attenuates HO progression. A TGF-β neutralizing antibody (1D11; R&D Systems, Minneapolis, MN) or vehicle antibody of an identical IgG complex lacking any TGF-β-binding capabilities (13C4; R&D Systems, Minneapolis, MN) was injected in the ATP-induced HO mice three times a week from the day of ATP (inflammatory

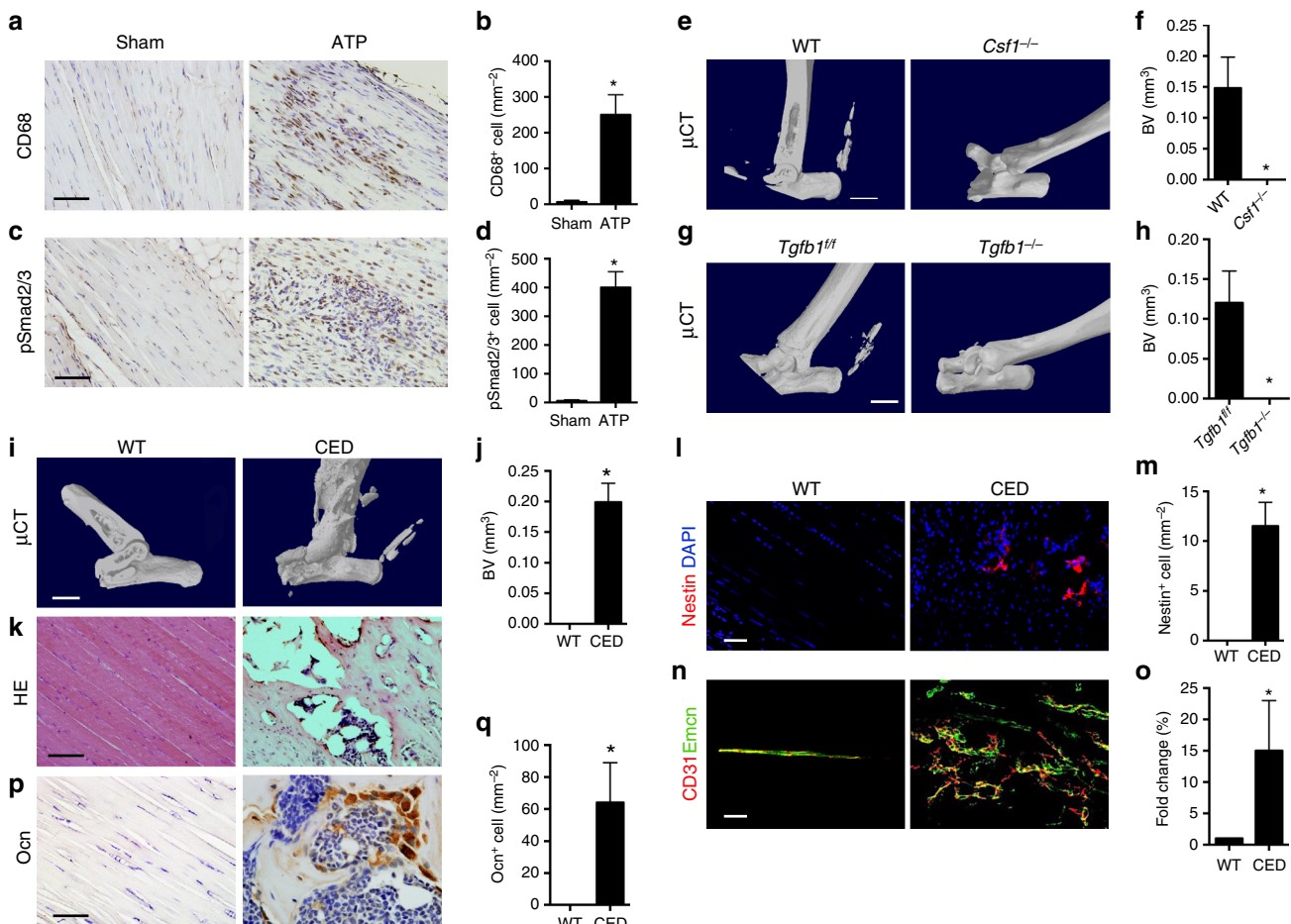

**Fig. 3** Macrophages produce TGF-β to initiate HO. **a–h** High levels of active TGF-β were associated with increased macrophage four days after HO induction by ATP. **a, c** Immunostaining and **b, d** quantitative analysis of CD68 and pSmad2/3-positive cells in Achilles tendon two days after sham operation or ATP. Scale bar, 50 μm. $n = 8$ per group. **e** Micro CT images and **f** quantitative analysis of HO bone volume in the Achilles tendons (sagittal view) in WT or $CSF1^{-/-}$ mice 6 weeks after ATP. Scale bar, 2 mm. $n = 8$ per group **g** Micro CT images and **h** quantification of HO bone volume in the Achilles tendons (sagittal view) in $Tgfb1^{flox/flox}$ or $LysM$-$cre$::$Tgfb1^{flox/flox}$ mice 6 weeks after ATP. Scale bar, 2 mm. $n = 8$ per group. **e–q** Transgenic expression of active TGF-β by Col I promoter induces HO. **i** Micro CT images of the Achilles tendon (sagittal view) of 4-month-old CED mice and WT littermates and **j** quantitative analysis of HO bone volume in Achilles tendon. Scale bar, 2 mm **k** H&E staining of Achilles tendon of 4-month-old CED mice and WT littermates. Scale bar, 50 μm. **l** Nestin⁺ (red) cells and **m** quantification in the ectopic bone marrow of CED and WT mice. Scale bar, 50 μm. Blue indicates DAPI staining of nuclei. **n** CD31⁺ (red) and Emcn⁺ (green) cells in the ectopic bone marrow. Scale bar, 100 μm. Yellow indicates type H vessels. **o** Quantification of the fold change of type H vessels in CED mice compared to that of WT littermates. **p** Immunostaining and **q** quantification of Ocn⁺ cells of in ectopic bone marrow in Achilles tendons of 4-month-old CED mice and WT littermates. Scale bar, 20 μm. All data are shown as the mean ± s.d. $n = 22$ per group. *$p < 0.05$ as determined by unpaired Student's $t$-test

stage = d1), 3 weeks (chondrogenesis stage = w3), 6 weeks (osteogenesis stage = w6), or 12 weeks (maturation stage = w12) post ATP for 3 weeks. The mice were killed 15 weeks after ATP. HO formation was significantly mitigated with injection of 1D11 antibody in d1, w3, and w6 groups relative to control antibody-treated mice (Fig. 4a–c). However, injection of 1D11 antibody at w12 post-surgery did not significantly reduce HO formation (Fig. 4a–c).

To evaluate the changes in signaling pathways for HO progression, experiments were repeated with sacrifice of mice 9 weeks after ATP. Immunostaining showed that the number of pSmad2/3⁺ cells was significantly decreased in HO bone marrow in mice with injection of 1D11 from day 1 (D1), week 3 (W3), and week 6 (W6) relative to control antibody-injected mice (Fig. 4d, e). Similar results were obtained in regards to active TGF-β concentration in serum (Fig. 4f). The numbers of Nestin⁺ cells, type H vessels, and Ocn⁺ osteoblasts were also significantly decreased in HO bone marrow with injection of 1D11 from D1,

W3, and W6 groups relative to vehicle-injected mice (Fig. 4g–l). In addition, we injected 3-month-old CED mice intraperitoneally with 1D11 or vehicle daily for 4 weeks. Analysis of Achilles tendons by micro CT showed that injection of TGF-β neutralizing antibody resulted in no HO formation (Supplementary Fig. 3). Collectively, these results indicate that inhibition of TGF-β signaling activity at the stages of inflammation, chondrogenesis, or osteogenesis attenuates HO progression.

**TGF-β antibody attenuates progression of BMP-induced HO.** Elevated BMP signaling induces HO of FOP patients and animal models. We then examined the effect of TGF-β antibody treatment on BMP-induced HO mouse model. BMP-2/Gelatin scaffolds were implanted in the hamstring muscles to generate HO (BMP-2/Gelatin Implantation model, BGI)[37]. Heterotopic bone was formed by 2 weeks after implantation and enlarged by 4 weeks (Supplementary Fig. 4a, b). H&E and SOFG staining

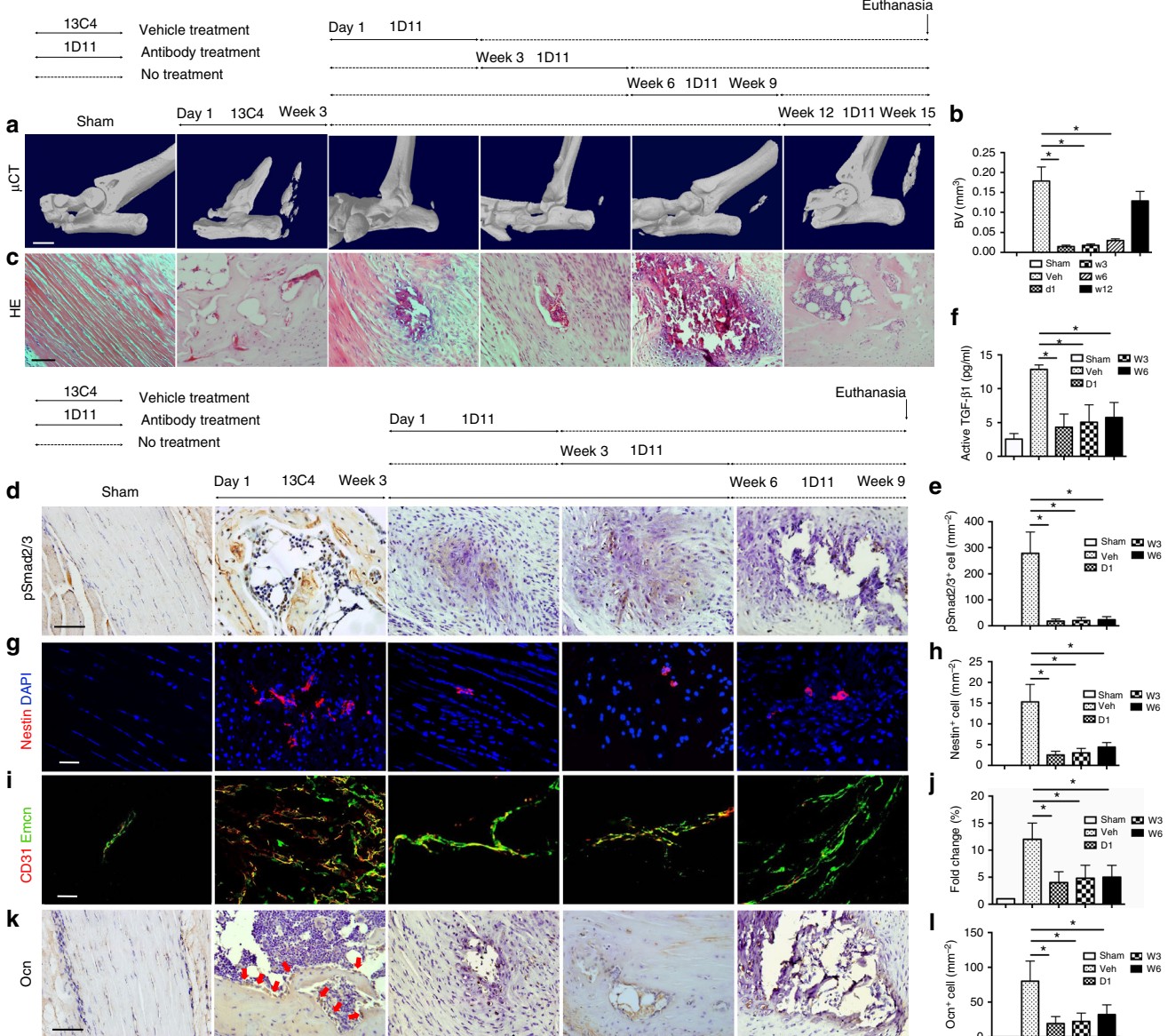

**Fig. 4** Systemic injection of TGF-β neutralizing antibody reduces ectopic bone formation and type H vessel formation. **a–c** Mice were treated with 5 mg/kg body weight of the TGF-β neutralizing antibody 1D11 three times a week for 3 weeks from 1 day (D1), 3 weeks (W3), 6 weeks (W6), and 12 weeks (W12) after ATP and analyzed 15 weeks after ATP or sham surgery. **a** Micro CT images of the Achilles tendon (sagittal view) of mice and **b** quantitative analysis of bone volume of heterotopic bone determined by μCT analysis. Scar bar, 2 mm. **c** H&E staining of ectopic bone of Achilles tendons. Scale bar, 50 μm. **d–l** Mice were treated with 5 mg/kg body weight of 1D11 three times a week for 3 weeks from 1 day (D1), 3 weeks (W3), and 6 weeks (W6) after ATP and analyzed 9 weeks after ATP or sham surgery. **d** Immunostaining and **e** quantification of pSmad2/3$^+$ cells in ectopic bone marrow. Scale bar, 50 μm. **f** Active TGF-β in serum determined by ELISA. **g** Nestin$^+$ (red) cells in the ectopic bone marrow and **h** quantification. Scale bar, 50 μm. Blue indicates DAPI staining of nuclei. **i** CD31$^+$ (red) and Emcn$^+$ (green) cells in the ectopic bone marrow. **j** Quantification of the fold change of Type H vessels in 1D11-treated ATP mice normalized to that of Sham mice. Scale bar, 100 μm. Yellow indicates type H vessels. **k** Immunostaining and **l** quantification of Ocn$^+$ cells of in ectopic bone marrow. Red arrow shows Ocn$^+$ cells. Scale bar, 50 μm. All data are shown as the mean ± s.d. $n = 8$ per group. *$p < 0.05$ as determined by one-way ANOVA

showed calcified cartilage formed at 2 weeks post-implantation and a fully developed ectopic bone with marrow at 4 weeks (Supplementary Fig. 4c, d) suggesting a mixed stage of chondrogenesis and osteogenesis at week 2 and maturation stage at week 4. The numbers of TRAP$^+$ cells, pSmad2/3$^+$ cells, and Ocn$^+$ osteoblasts were increased in the heterotopic bone at 2 weeks and decreased to the level of sham-operated mice by 4 weeks after surgery (Supplementary Fig. 4e–j). Collectively, this suggests that TGF-β activity is also present following HO induction by elevated BMP signaling.

We then examined whether TGF-β plays a direct role in HO progression of BGI mice. BGI-induced HO mice were injected with 1D11 antibody from day 1 (D1), week 2 (W2), or week 4 (W4) post-implantation for 2 weeks and killed at 8 weeks. HO formation in hamstring muscles was abolished with injection of 1D11 from day 1 post-implantation. HO bone formation was also significantly decreased when 1D11 was injected after W2, while no significant reduction of HO bone volume after W4 relative to control antibody-treated mice by μCT analysis and H&E was found (Fig. 5a–c). To examine the cellular mechanism, BGI HO

mice injected with 1D11 from day 1 (D1) or week 2 (W2) post-implantation for 2 weeks were also killed 4 weeks post-implantation. There were very few numbers of pSmad2/3+ cells in the HO site of all groups (Fig. 5d, e). Similar to ATP HO mice injected with 1D11 antibody, the number of Nestin+ cells was significantly decreased in 1D11 groups treated from either D1 or W2 relative to controls (Fig. 5f, g). Moreover, the formation of type H vessels and number of Ocn+ osteoblasts were significantly reduced in mice injected with 1D11 relative to control antibody-injected mice, indicating reduced bone formation (Fig. 5h–k).

We also analyzed the mobility of the mice by comparing the gait of ATP or BGI mice treated with vehicle or 1D11 from osteogenesis stage with that of sham control mice. The resulting footprint patterns were assessed quantitatively by two measurements: stride length and front footprint/hind footprint overlap. The results displayed a similar stride length and uniformity in step alternation after HO induction treated with vehicle or 1D11 with no significances relative to controls (Supplementary Fig. 5). Using a mouse model of FOP in which a constitutively active mutant form of *ALK2* (*caALK2*) is expressed upon injection of

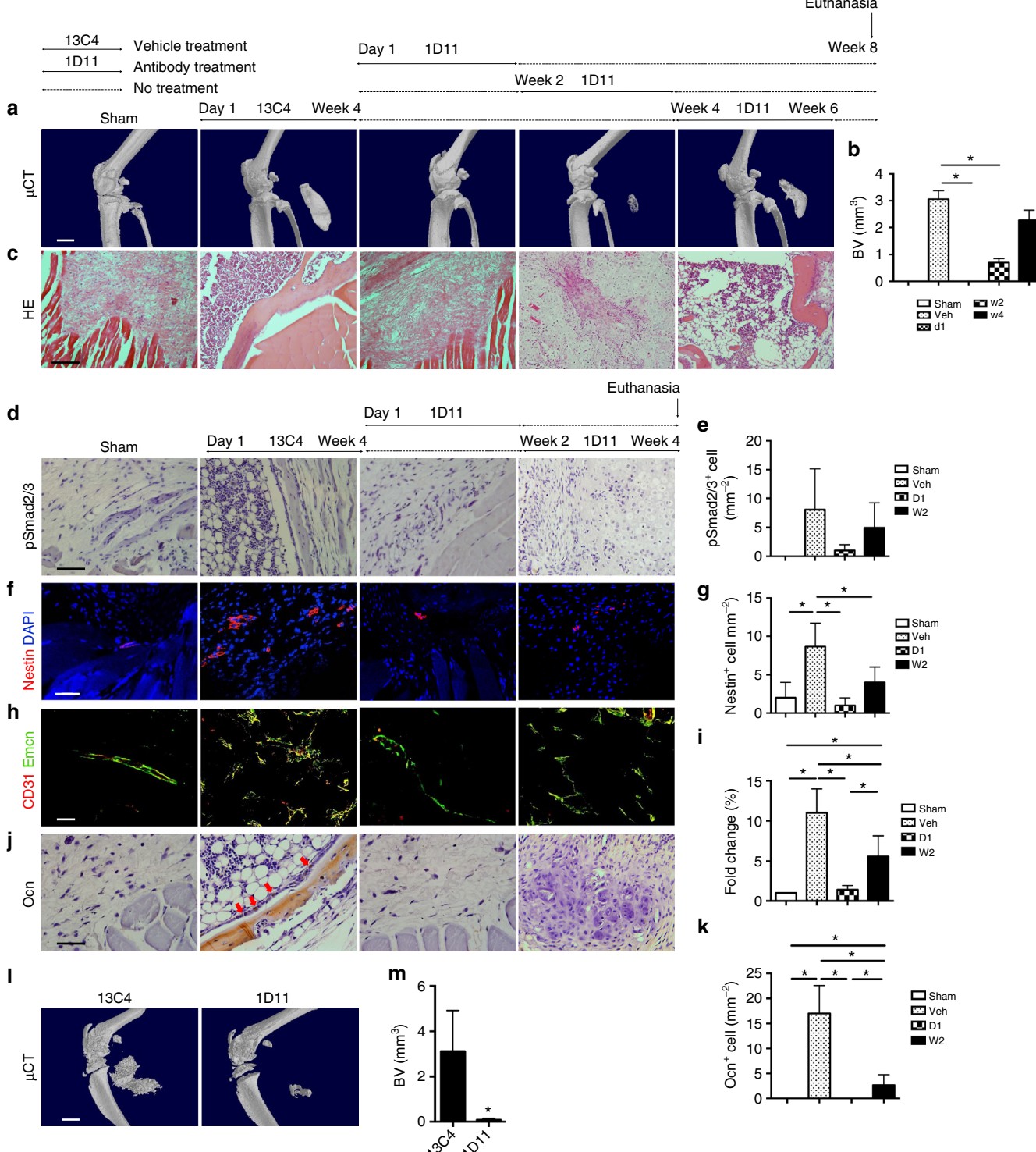

Adenovirus Cre (Ad.Cre) and cobra venom factor[38], we found injection of 1D11 also significantly decreased HO progression (Fig. 5l, m). Taken together, our data demonstrate that progression of HO by active TGF-β is a common pathomechanism in both acquired HO and FOP.

**Nestin+ MSPCs are involved in angiogenesis for HO formation**. To understand the cells increased by TGF-β for ectopic bone formation, we analyzed Nestin+ cells during HO progression using BGI-induced HO in a transgenic *Nestin-GFP* mouse model[39]. CD45−GFP+ Nestin cells in the HO lesions were isolated by flow cytometry 4 weeks post-BGI. Co-expression of the leptin receptor (LepR), a recognized marker for mouse bone marrow MSPCs[40] was detected in 85% of HO marrow Nestin-GFP+ cells (Fig. 6a). About 80% of the GFP+LepR+ cells also expressed Sca-1 or CD105, cell surface markers of MSPCs[17,19], whereas only 2% of the GFP+LepR+ cells expressed CD31 (Fig. 6b, Supplementary Fig. 6). By contrast, 90% of the GFP+LepR− cells were CD31 positive, Sca-1, and CD105 negative (Fig. 6c, Supplementary Fig. 6). GFP+ cells were capable of both osteogenesis as indicated by alizarin red staining (Fig. 6d) and angiogenesis as indicated by tube formation using in vitro Matrigel assay (Fig. 6e), indicating that Nestin+ cells in HO marrow are involved in both osteogenesis and angiogenesis.

We next generated tamoxifen-inducible *Nestin-creERT2::R26R-EYFP* mice to trace Nestin+ cells during HO progression after HO induction by ATP for 3 weeks and 6 weeks. More than 90% of cartilage cells were derived from the Nestin+ lineage cells 3 weeks after ATP (Fig. 6f, g). Co-immunostaining of CD31/YFP and Emcn/YFP revealed about 70−86% of type H vessel were formed by Nestin lineage cells 6 weeks after ATP during HO development (Fig. 6h, i). Tendon residing Scx+ cells have been proved as precursors for HO in an FOP mouse model[9]. To test if Scx+ cells in tendon are also the main cell source for acquired HO in ATP mice, we generated a tamoxifen-inducible *Scx-creERT2::R26R-EYFP* mouse model by crossing *Scx-creERT2* mice and *R26R-EYFP* mice. Only less than 2% Scx+ cells contribute to cartilage formation and less than 5% to vessel formation (Supplementary Fig. 7a−d). It is plausible that different cell cohorts are recruited in genetic vs non-genetic HO.

We also generated TGF-β type II receptor (*Tgfbr2*)-inducible knockout mice (*Nestin-creERT2::Tgfbr2^flox/flox^*). When *Nestin-creERT2::Tgfbr2^flox/flox^* mice underwent ATP or BGI, the mice were injected with tamoxifen three times weekly to specifically delete *Tgfbr2* (*Tgfbr2^−/−^*) in the Nestin+ cells. Interestingly, no HO was observed in *Tgfbr2^−/−^* mice in either ATP or BGI models, while HO was evident in *Nestin-creERT2::Tgfbr2^flox/flox^* vehicle-injected control mice (*Tgfbr2^flox/flox^*) (Fig. 7a−f). We also used *Nestin-creERT2* mice (*Cre*) treated with tamoxifen as a control to exclude tamoxifen as a confounder and found ectopic bone formation in *Cre* mice after ATP or BGI. There were no significant differences between tamoxifen-treated *Cre* mice and

*Tgfbr2^flox/flox^* mice (Supplementary Fig. 8a−d). Moreover, Nestin+ cells were not found in Achilles tendons or rarely found in hamstring muscles in ATP or BGI *Tgfbr2^−/−^* mice compared to HO marrow in corresponding *Tgfbr2^flox/flox^* HO mice, respectively (Fig. 7g−j). Similarly, type H vessels in Achilles tendons or hamstring muscles were significantly decreased in *Tgfbr2^−/−^* mice (Fig. 7k−n). These data further demonstrate active TGF-β increases Nestin+ cells for angiogenesis and drives ectopic bone formation, inhibition of which attenuates HO progression.

**Discussion**
HO was first described more than 300 years ago, however, we still have limited knowledge about acquired HO with no effective therapy, leaving surgical excision as the primary treatment for mature HO[41]. Our data uncovered the nature of HO progression as an excessive activation of TGF-β in recruitment of MSPCs for osteogenesis in coupling with type H vessel formation. Systemic injection of TGF-β neutralizing antibody effectively attenuated HO progression in multiple different HO animal models. TGF-β has a broad spectrum of function including inflammation, cell migration, chondrogenesis, angiogenesis, epithelial mesenchymal transition (EMT), EndMT and remodeling of the new [ECM[13,42]. Previous studies have shown that TGF-β is activated after injuries and is required in all phases of chondrogenesis, mesenchymal condensation, chondrocyte proliferation, ECM deposition, and finally terminal differentiation[43,44]. Particularly, spatiotemporal activation of matrix latent TGF-β maintains homeostasis of skeletal tissues, and high concentrations of active TGF-β leads to abnormal bone formation[17,19,42,45]. Disruption of the precise activation of TGF-β leads to a wide variety of diseases, and inhibition of TGF-β activity has been a common approach[17,18,21–25,46–55]. At the early stage of HO, immune cells produce abundant TGF-β and degradation of ECM or LAP by inflammation may contribute to TGF-β activation[56]. During the osteogenesis stage, TGF-β is activated by osteoclastic bone resorption[17]. Blockage of active TGF-β effectively mitigates HO progression at broad window during HO progression in different animal models likely in a dose-dependent fashion. It did not eliminate the already formed HO. However, given longer treatment time, the ectopic bone may be resorbed by osteoclastic bone resorption as inhibition of TGF-β activity would not affect osteoclasts.

HO progression is an energy consuming process and requires blood vessels to transport nutrients, oxygen, glucose, amino acids, and minerals for osteogenesis and removing osteoclast resorptive waste. We found that Nestin+ cells give rise to type H vessels during the progression of HO in coupling with osteogenesis. Ablation of *Tgfbr2* in Nestin lineage cells caused a marked reduction in the formation of both blood vessels, cartilage, and further ectopic bone formation. Nestin+ cells are heterogeneous populations providing the precursors to mesenchymal and endothelial lineages[30–33]. It is plausible that Nestin lineage cells are recruited in the early stage of HO as part of the inflammatory

**Fig. 5** TGF-β neutralizing antibody attenuates HO induced by BMP-2/gelatin implantation and FOP mice. **a–c** Mice were treated with 5 mg/kg body weight of the TGF-β neutralizing antibody 1D11 three times a week for 2 weeks from 1 day (D1), 2 weeks (W2), and 4 weeks (W4) after BGI and analyzed 8 weeks after BGI or sham surgery. **a** micro CT images and **b** quantitative analysis of bone volume of heterotopic bone in hamstring muscles of mice. Scar bar, 4 mm. **c** H&E staining of ectopic bone in hamstring muscles. Scale bar, 100 μm. **d–k** Mice were treated with 5 mg/kg body weight of 1D11 three times a week for 2 weeks from 1 day (D1) and 2 weeks (W2) after BGI and analyzed 4 weeks after BGI or sham surgery. **d** Immunostaining and **e** quantification of pSmad2/3+ cells in ectopic bone marrow. Scale bar, 50 μm. **f** Nestin+ (red) cells in the ectopic bone marrow and **g** quantification. Scale bar, 50 μm. Blue indicates DAPI staining of nuclei. **h** CD31+ (red) and Emcn+ (green) cells in the ectopic bone marrow and **i** quantification of the fold change of type H vessels normalized to that of Sham mice. Scale bar, 100 μm. Yellow indicates type H vessels. **j** Immunostaining and **k** quantification of Ocn+ cells of in ectopic bone marrow. Scale bar, 50 μm. **l**, **m** Ad.Cre and cobra venom factor-injected *caALK2* transgenic mice were treated with vehicle or 1D11 three times a week for 3 weeks. **l** Micro CT images of hamstring muscles after vehicle or 1D11 treatment and **m** quantitative analysis of bone volume. Scale bar, 4 mm. All data are shown as the mean ± s.d. $n = 8$ per group. **b**, **e**, **g**, **i**, **k** *$p < 0.05$ as determined by one-way ANOVA. **m** *$p < 0.05$ as determined by unpaired, two-tailed Student's *t*-test

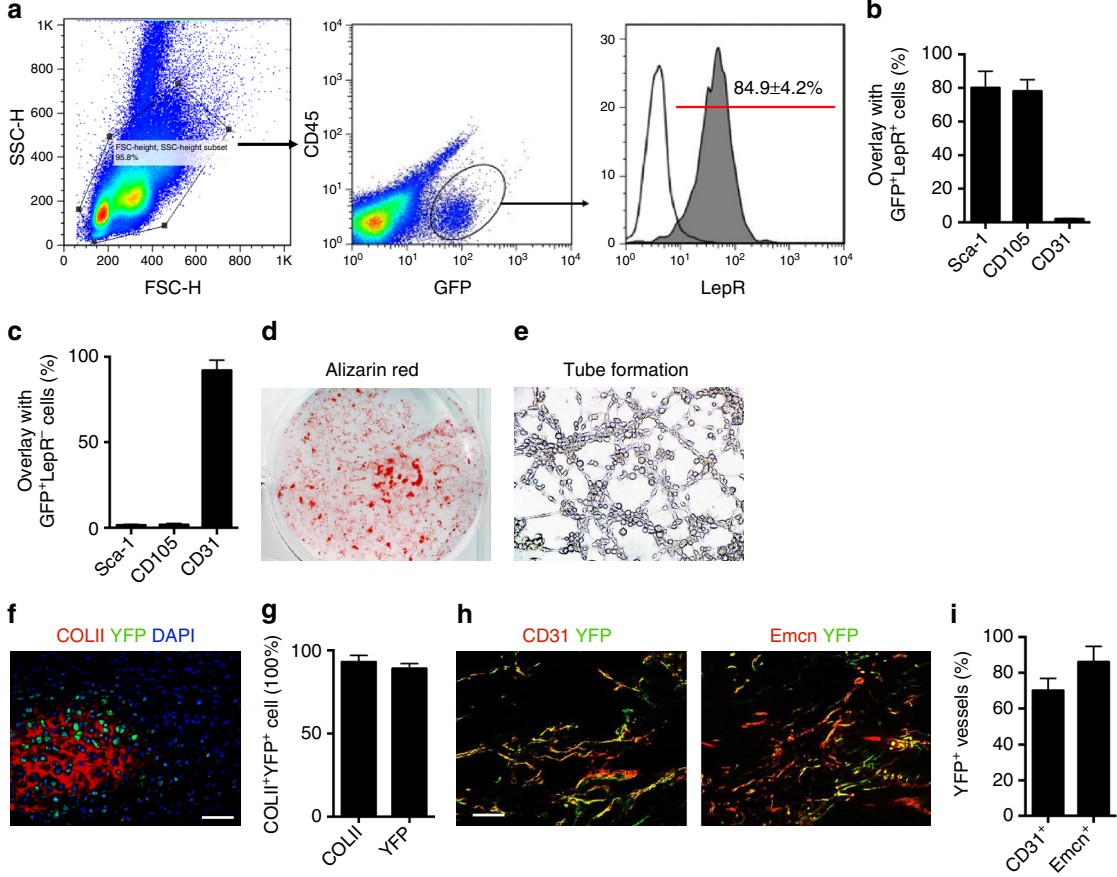

**Fig. 6** Nestin[+] cells at the injured site are predominantly MSPCs. **a** The percentage of the CD45[-]GFP[+] cells that express LepR. $n = 16$. **b** The percentages of the GFP[+]LepR[+] cells or **c** GFP[+]LepR[-] cells that express Sca-1, CD105, or CD31, respectively. **d** Alizarin red staining of GFP[+] cells cultured in osteogenic differentiation medium for 14 days and **e** tube formation of GFP[+] cells by Matrigel assay. **f** Immunostaining of COLII[+] cells (red) and YFP[+] cells (green) and **g** quantitation of *Nestin-creERT2::EYFP[flox/flox]* mice 3 weeks post ATP. Scale bar, 50 μm, $n = 8$ per group. **h**, left: CD31[+] cells (red), YFP[+] cells (green), and right: Emcn[+] cells (red), YFP[+] cells (green), and **i** quantitation in ectopic bone marrow of *Nestin-creERT2::EYFP[flox/flox]* mice 6 weeks post ATP. Scale bar, 100 μm, $n = 8$ per group. All data are shown as the mean ± s.d

response to form endothelial vessels[12] by Nestin[+] endothelial precursor sub-population and the Nestin[+] mesenchymal precursor sub-population contribute to the formation of heterotopic cartilage. Our study also outlines the key signaling pathways as targets for treatment at various stages of HO progression. TGF-β antibody effectively blocked the progression of HO if applied at either inflammation, chondrogenesis, or osteogenesis stage, further implicating the fundamental role of TGF-β in HO progression. The present study provides a potential treatment for HO in the stage when osteogenesis already occurs. The ectopic bone formation seen in POH, a process of intramembranous bone formation, may not involve TGF-β signaling.

BMP signaling induces commitment of progenitor cells/endothelial cells to osteoblasts[9,57], and more importantly, high levels of active TGF-β initiates HO and is associated with the recruitment of these cells for ectopic bone formation and angiogenesis. FOP patients with constitutive BMP signaling could induce HO spontaneously, but vigorous ossification occurs with additional stimuli, such as inflammation or injuries[58,59]. Additionally, the severity of symptom and/or disease progression among FOP patients are not equal. A recent study showed high levels of TGF-β signaling in fibroblasts of FOP patients[60]. The *caALK2* mouse used in present study aims to introduce a hyperactive BMP receptor[10,38]. The interpretation of 1D11 antibody inhibition of HO in *caALK2* mice is that 1D11 antibody neutralizes TGF-β ligands essential for HO initiation at the inflammation stage. The

transgenic mouse line used here bears mutant *ALK2[Q207D]* similar to human *ALK2[R206H]*. However, it is a strong constitutively activating mutation that induces much higher levels of pSmad1/5/8 signaling compared the much more mildly activating *R206H* mutation. Thus, this mouse model could not mimic all aspects of the human conditions. It needs a combination of trauma and cobra venom or cardiotoxin to initiate HO; however, a trivial injury could lead to vigorous HO in human FOP patients.

In the present study, we found that inciting events such as injuries to the Achilles tendons or BMP-2/Gelatin implantation in hamstring muscles reliably induced HO and increased active TGF-β levels throughout HO progression. Previous research that has been explored for HO aimed to inhibit inflammation with NSAIDs[61,62], BMP signaling with BMP inhibitors[9,63], or chondrogenesis with nuclear retinoic acid receptor-γ agonists[37]. However, we still do not have an effective therapy for HO including inhibition of BMP signaling[64,65]. In the present study, we found that the inhibition of TGF-β activity successfully mitigates HO at different stages of HO. Therefore, inhibition of TGF-β signaling could be a potential target for a medical intervention to treat debilitating, painful, and recurrent acquired HO and FOP.

## Methods

**Human subjects**. The study was approved by Johns Hopkins University and Shanghai Sixth People's Hospital internal review board. As human samples were

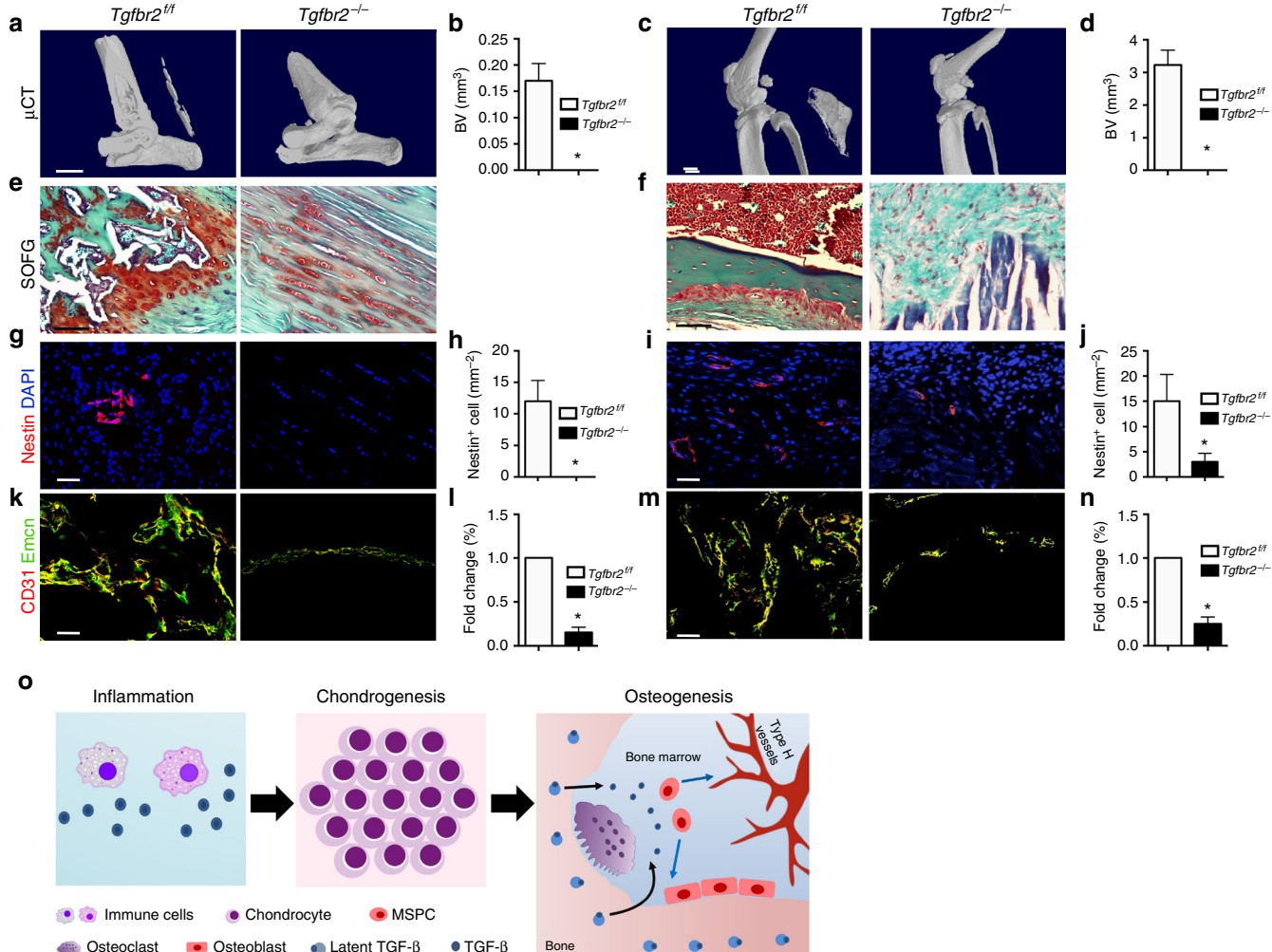

**Fig. 7** Inducible knockout of *Tgfbr2* in Nestin$^+$ cells attenuates HO formation in ATP and BGI mice. **a**, **c** Micro CT images and **b**, **d** quantifications of **a**, **b** Achilles tendons or **c**, **d** hamstring muscles of *Nestin-creERT2::Tgfbr2$^{flox/flox}$* mice 2 months treated with vehicle or tamoxifen after undergoing ATP or BGI surgery, respectively. Scale bar, 2 mm. **e**, **f** SOFG staining of **e** Achilles tendons or **f** hamstring muscles of *Tgfbr2$^{flox/flox}$* and *Tgfbr2$^{-/-}$* mice 2 months after ATP or BGI, respectively. Scale bar, 50 μm **g**, **i** Nestin$^+$ (red) cells in the ectopic bone marrow and **h**, **j** quantifications of *Tgfbr2$^{flox/flox}$* and *Tgfbr2$^{-/-}$* mice after ATP or BGI, respectively. Scale bar, 50 μm. Blue indicates DAPI staining of nuclei. **k**, **m** CD31$^+$ (red) and Emcn$^+$ (green) cells in the ectopic bone marrow and quantifications of *Tgfbr2$^{flox/flox}$* and *Tgfbr2$^{-/-}$* mice after ATP or BGI. Scale bars, 100 μm. Yellow indicates type H vessels. **a**–**n** $n = 8$ per group. All data are shown as the mean ± s.d. *$p < 0.05$ as determined by two-tailed, unpaired Student's *t*-test. **o** Hypothetical diagram of HO formation. After HO induction, abundant immune cells are infiltrated in the injury sites and produce TGF-β. Next, the chondrocytes proliferate and undergo hypertrophy and calcification. Finally, osteoclasts resorb calcified cartilage and free TGF-β from latent protein and diffuse to the ectopic bone marrow cavity. Active TGF-β recruits MSPCs to form type H vessels and for osteoblastic differentiation

de-identified excess pathology tissues, the review board granted an exemption from the requirement of obtaining written informed consent from individuals. For the pathological specimens, we collected acquired HO identified radiographically from 18 patients (13 male and 5 female, previously healthy, nonsmoking individuals; age ranging from 26 to 47 years) who had previously sustained an elbow fracture that was treated by internal fixation and had returned for clinical evaluation secondary to HO. Osteogenesis (9 patients, 3–4 months after surgery) or maturation stage HO (9 patients, from 14–16 months after surgery; Bony ankylosis of elbow) was defined based on the number of months since their fixation surgery. We recruited an additional 18 patients with HO after CNST (nine male and nine female previously healthy, nonsmoking individuals; age ranging from 24 to 55 years; eight after traumatic brain injuries and ten after spinal cord injuries), nine of which were from osteogenesis stage (4–5 months after injuries; three after traumatic brain injuries and six after spinal cord injuries) and nine from maturation stage (14–16 months after injuries; five after traumatic brain injuries and four after spinal cord injuries). The healthy muscles were from 9 age-matched and gender-matched patients who had undergone total elbow replacement to serve as baseline controls. For the serum samples, a second prospective cohort was studied, after institutional review board approval was obtained from Shanghai Sixth People's Hospital, which included eight patients (six male and two female previously healthy, nonsmoking individuals; age ranging from 22 to 39 years) with pain or decreased elbow movement and found by radiography to have HO who provided informed consent for the study. All had a

history of elbow fracture status post internal fixation. Blood specimens (20 mm per person) were collected 3–4 and 14–16 months after internal fixation. Twenty millimetre blood from eight age-matched and gender-matched healthy subjects were used as baseline controls. All the serum samples were fasting samples, collected in the morning in the orthopedic outpatient clinic. Specimens were processed immediately and serum samples were stored in -80°C freezer. The serum specimens were processed for ELISA examination. Exclusion criteria for both studies included patients who received any HO treatments, such as NSAIDs, local irradiation, or surgery either before or during the course of the study.

**Mice**. We purchased 3-month-old C57BL/6J (WT, Stock number: 000664) mice from the Jackson Laboratory. For ATP mouse model, 10-week-old male mice were anesthetized by ketamine and xylazine. A 27-gauge needle was punctured into the Achilles tendon body from the lateral aspect percutaneously and this process was repeated five times at different parts of Achilles tendon body for each mouse. For sham operation, the needle was punctured through the skin without touching the Achilles tendons.

For the BGI mouse model, 3% (g/ml) gelatin (Sigma-Aldrich, G1890) solution was cross-linked with 0.1% (g/g) glutaraldehyde (Sigma-Aldrich, G5882) followed by lyophilization at −45 °C to prepare a gelatin sponge. The sponge was then cut into 1 mm thick discs ($d = 3$ mm) and coated with 1 μl aliquot containing 1 μg of

recombinant human BMP-2 (GenScript, Z02913) for each disc. Under general anesthesia [Ketamine (100 mg/kg) and Xylazine (15 mg/kg)], longitudinal skin incisions were made on the medial surface of thigh muscles in the left legs of 10-week-old male mice. An intramuscular pocket was created in the hamstring muscle microsurgically, and one BMP-2/Gelatin disc was placed in. The skin was then closed with 4.0 silk sutures.

For the antibody treatment experiments, 10-week-old ATP or BGI-operated male mice were intraperitoneally injected 13C4 (R&D Systems, Minneapolis, MN) from the day of ATP or BGI (controls) or 1D11 (R&D Systems, Minneapolis, MN) 5 mg/kg body weight three times a week from the day of ATP, 3 weeks, 6 weeks, or 12 weeks after ATP for 3 weeks or from the day of BGI, 2 weeks, or 4 weeks after BGI for 2 weeks. 1D11 is a monoclonal antibody that neutralizes 3 major active TGF-β isoforms (TGF-β1, -2, and -3), the known ligands for the TGF-β receptor kinase. It does not block the receptor or bind other ligands in the TGF-β superfamily, including activin or BMP[66]. The mice were killed 15 weeks (ATP) or 8 weeks (BGI) after operation. To further illustrate the changes in signaling pathways for HO progression, the experiments were repeated after killing the mice (1) 9 weeks after ATP with 1D11 injection from the day of ATP, 3 weeks, or 6 weeks for 3 weeks or (2) 4 weeks after BGI with 1D11 injection from the day of BGI or 2 weeks after BGI for 2 weeks. The mice with 13C4 injection from the day of operation for 3 weeks (ATP) or 2 weeks (BGI) were used as controls. Sham-operated mice without any treatment were used as baseline controls.

We purchased the 8-week-old Csf1[op] (Stock number: 000231), LysM-cre (Stock number: 004781), and Tgfb1[flox/flox] (Stock number: 010721) mouse strains from Jackson Laboratory. We generated Csf1[−/−] offspring and their wild-type littermates (WT) by crossing two heterozygote Csf1[op] strains. We generated LysM-cre:: Tgfb1[flox/flox] mice (Tgfb1[−/−]) by crossing LysM-cre mice with Tgfb1[flox/flox] mice. We performed ATP operations on 10-week-old WT, Csf1[−/−], Tgfb1[flox/flox], and Tgfb1[−/−] male mice, and killed them after 6 weeks for micro CT analysis.

CED mice were generated in our laboratory as previously described, in which the CED-derived TGF-β1 mutation (H222D) was specifically expressed by osteoblastic cells driven by a 2.3-kb COL1α 1 promoter[17]. Twenty-two 4-month-old male CED mice were used for analyzing spontaneous HO formation. Twenty-two 4-month-old male WT mice were used as controls. For the antibody treatment experiments, 16 3-month-old male CED mice were intraperitoneally injected 13C4 (8 mice) or 1D11 (8 mice) 5 mg/kg body weight daily for 4 weeks before being killed for micro CT analysis.

Nestin-GFP mice were provided by Dr Grigori Enikolopov at Cold Spring Harbor Laboratory[39]. At the age of 4 weeks, 16 Nestin-GFP male mice underwent BGI and were killed 4 weeks after operation. The ectopic bone was collected for cell sorting and flow cytometry analysis.

We purchased 8-week-old Nestin-creERT2 (Stock number: 016261) mice and R26R-EYFP (Stock number: 006148) from the Jackson Laboratory. Mice with floxed Tgfb2 (Tgfb2[flox/flox]) were obtained from the lab of H.L. Moses[67]. We generated Nestin-creERT2::R26R-EYFP mice by crossing Nestin-creERT2 mice with R26R-EYFP mice. We performed ATP operations on 10-week-old Nestin-creERT2:: R26R-EYFP male mice. Three days after surgery, we treated the mice with 80 mg/kg body weight of tamoxifen three times a week for 3 or 6 weeks and killed the mice at 3 or 6 weeks after surgery. Scx-creERT2 mice were generously provided by Dr. Ronen Schweitzer at Shriners Hospital for Children. We generated Scx-creERT2:: R26R-EYFP mice by crossing Scx-creERT2 mice with R26R-EYFP mice. We performed ATP operations on 10-week-old Nestin-creERT2::R26R-EYFP male mice. Three days after the surgery, we treated the mice with 80 mg/kg body weight of tamoxifen three times a week for 3 or 6 weeks and killed the mice at 3 or 6 weeks after surgery. We generated Nestin-creERT2::Tgfbr2[flox/flox] mice by crossing Nestin-creERT2 mice with Tgfbr2[flox/flox] mice. We performed ATP or BGI operations on 10-week-old Nestin-creERT2::Tgfbr2[flox/flox] male mice (16 mice for ATP and 16 mice for BGI). One day after the surgery, we treated the mice with either 80 mg/kg body weight of tamoxifen (eight for ATP and eight for BGI) or vehicle (eight for ATP and eight for BGI) three times a week for 9 weeks (ATP) or 4 weeks (BGI) and killed the mice 9 weeks after ATP or 4 weeks after BGI. Sixteen ATP or BGI-operated Nestin-creERT2 male mice (eight mice for each operation) injected with tamoxifen three times a week for 9 weeks (ATP) or 4 weeks (BGI) and killed the mice 9 weeks after ATP or 4 weeks after BGI were also used as controls.

Constitutively active ALK2 (caALK2) mice were provided by Dr Yuji Mishina at University of Michigan[38]. To induce expression of caALK2, adenoviral-Cre (Vector Biolabs, 1045; 1 × 10$^9$ PFU per mouse) and cobra venom factor (EMD/Millipore, 233552; 0.03 μg per mouse) were injected into the left hindlimbs of sixteen male mice at 4 weeks of age. The mice were then injected with 13C4 (eight mice) or 1D11 (eight mice) 10 mg/kg body weight three times a week for 3 weeks from 3 days after HO induction and killed at the age of 7 weeks.

All animals were maintained in the Animal Facility of the Johns Hopkins University School of Medicine. The experimental protocols were reviewed and approved by the Institutional Animal Care and Use Committee of the Johns Hopkins University, Baltimore, MD, USA.

**Specimen collection.** Mice were killed by carbon dioxide ($CO_2$) inhalation and perfusion fixed with 10% buffered formalin via the left ventricle for 5 min. Then we dissected the ankles with Achilles tendons and fixed the specimens in 10% buffered formalin for 24 h, decalcified in 10% Ethylenediaminetetraacetic acid (EDTA, VWR, 0105) (pH 7.4) for 14 days, and embedded in paraffin, Optimal Cutting

Temperature Compound (O.C.T. compound, VWR, 25608-930,) for 3 days and embedded in matrix containing 8% Gelatin (Sigma-Aldrich, G1890), 20% sucrose (Sigma-Aldrich, S9378), and 2% Polyvinylpyrrolidone (Sigma-Aldrich, PVP40) at −80 °C adjusted from previous protocol[68]. The majority of analyses were in paraffin-embedded specimens, while detection of Nestin, CD31, and Emcn was more optimal in frozen specimens.

**Histochemistry, immunohistochemistry, and histomorphometry.** The blocks were sectioned at 4 μm or 80 μm (for CD31 and Emcn immunofluorescent staining) intervals using a Microm cryostat (for frozen blocks) or a Paraffin Microtome (for paraffin blocks). We processed 4-μm-thick sections of bone for H&E staining and safranin o (Sigma-Aldrich, S2255) and fast green (Sigma-Aldrich, F7252) staining. Trap staining was processed following the manufacturer's protocol (Sigma-Aldrich, 387A-1KT), followed by counterstaining with Methyl Green (Sigma-Aldrich, M884). We performed immunohistochemistry and immuno-fluorescence analysis as described previously[69]. Both de-waxed paraffin sections and frozen sections were heated to 99 °C for 20 min in Target Retrieval Solution (Dako, S1699) for antigen retrieval, and were rehydrated. After washing three times with PBS, the tissue sections were incubated with primary antibodies to human/ mouse pSmad2/3 (Santa Cruz Biotechnology Inc., sc-11769, 1:50), human/mouse PDGF-BB (Abcam, ab21234, 1:50), human CD73 (Abcam, ab54217, 1:100), human CD90 (Abcam, EPR3133, 1:100), mouse osteocalcin (Abcam, ab93876, 1:200), mouse nestin (Aves Labs, Inc., 1:300, lot NES0407), mouse CD31 (Abcam, ab28364, 1:100), mouse endomucin (Santa Cruz, V.7C7, 1:50), mouse CD68 (Abcam, ab31630, 1:100) and mouse GFP (Abcam, ab290, 1:200) overnight at 4 °C in a humidified chamber. The sections were washed three times with Tris-buffered saline. For immunohistochemical staining, we incubated slides with secondary antibodies in blocking solution for 1 h at room temperature and subsequently used Chromogenic Substrates (Dako, K3468) to detect the immunoactivity, followed by counterstaining with hematoxylin (Sigma-Aldrich, H9627). For immuno-fluorescence staining, we continued to use secondary antibodies conjugated with fluorescence at room temperature for 1 h, while avoiding light and mounted on slides with ProLong Gold Mounting Reagent with DAPI (Life Technologies, P36935). We used isotype-matched controls, such as polyclonal rabbit IgG (R&D Systems, AB-105-C), polyclonal goat IgG (R&D Systems, AB-108-C), and mono-clonal rat IgG2A (R&D Systems, 54447) under the same concentrations and conditions as negative controls. We used a Zesis 780 confocal microscope or an Olympus DP71 microscope for imaging samples. For the human specimens, five sequential sections per stain were analyzed. Anatomic landmarks to ensure com-parability included the presence of bone marrow and bone matrix. For the animal studies, serial sagittal sections of the HO lesions were obtained. We counted the numbers of positively stained cells in five random visual fields in five sequential sections per mouse in each group and normalized them to the number per milli-meter of adjacent bone surface (for TRAP staining quantification) or per square millimeter in HO area. We conducted the Quantitative analysis with OsteoMea-sureXP Software (OsteoMetrics, Inc.). For type H vessel quantification, we calcu-lated the area of yellow color in the whole HO site of each slide in three sequential sections per mouse in each group and normalized to that of sham mice (set to 1). For quantification of Nestin lineage cells, we used Nestin-creERT2::R26R-EYFP mice to trace Nestin lineage cells and all YFP$^+$ cells were considered as Nestin lineage cells. For chondrocytes quantification, we calculated all the cells in red (COLII$^+$) area and considered them as COLII$^+$ chondrocytes. The YFP$^+$COLII$^+$ cells were considered as chondrocytes that derived from Nestin$^+$ cells. Similarly, YFP$^+$CD31$^+$ and YFP$^+$Emcn$^+$ were considered as vessel cells derived from Nestin$^+$ cells. Quantifications were performed using ImageJ 1.48u4 software.

**Serum TGF-β1 and PDGF-BB analysis.** We determined the concentration of active TGF-β1 in the serum using the TGF-β1 ELISA Development kit (human: R&D Systems, DB100B; mouse: R&D Systems, MB100B) and PDGF-BB using PDGF-BB ELISA Development kit (human: R&D Systems, DBB00; mouse: R&D Systems, MBB00) following the manufacturer's instructions.

**MicroCT analysis.** Achilles tendons with calcaneus and lower tibia (for ATP and CED mouse models) or hindlimbs (for BGI and caALK2 mouse models) from mice were fixed overnight in 10% formalin and analyzed by high-resolution μCT (Skyscan1172). The scanner was set at a voltage of 60 kV and a resolution of 18 μm per pixel. The images were reconstructed, analyzed for HO bone volume, and visualized by NRecon v1.6, CTAn v1.9, and CTVol v2.0, respectively.

**Cell sorting and flow cytometry analysis.** For the analysis or sorting of Nestin$^+$ cells, we collected ectopic bone of BGI-induced HO Nestin-GFP mice after they were killed, removed all surrounding tissues, and then crushed the ectopic bone in ice-cold PBS. We digested whole bone with collagenase at 37 °C for 20 min to obtain single cell suspensions. After filtration with RBC lysis with commercial ACK lysis buffer (Quality Biological, 10128) and washing with 0.1% BSA in PBS, we counted the cells and incubated equal numbers of cells for 45 min at 4 °C with primary antibody APC/Cy7 anti-mouse CD45 Antibody (BioLegend, 103115, 1:100). The cells were then sorted according to side scatter and GFP expression after negative selection of CD45. For analyzing GFP$^+$ cells, single cell suspension

was first gated to select GFP$^+$LepR$^+$ and GFP$^+$LepR$^-$cells, and then CD45$^-$Sca1$^+$, CD45$^-$CD105$^+$, or CD45$^-$CD31$^+$ cells. Fluorescence-activated cell sorting (FACS) was performed using a five-laser BD FACS and FACSDiva. Flow cytometric analyses were carried out using a FACSCalibur flow cytometer and CellQuest software (Becton Dickinson). Other primary antibodies used were APC-conjugated anti-Sca1 (BioLegend, 122511, 1:100), APC-conjugated anti-CD31 (BioLegend, 102409, 1:100), and APC-conjugated anti-CD105 (BioLegend, 120413, 1:100). Antibody against Leptin Receptor was purchased from R&D (R&D, AF497, 2.5 ng/10$^3$ cells), followed by incubation with Cy3-conjugated secondary antibody (Abcam, ab6949, 1:1000).

**Characterization of nestin$^+$ cells.** Sorted GFP$^+$CD45$^-$ cells were used for investigating the osteogenic differentiation and tube formation of Nestin$^+$ cells. For osteogenic differentiation, the cells were seeded at a density of $5 \times 10^3$ cm$^2$ with α Minimum Essential Medium (αMEM, Corning, 10-022-CV) supplemented with 10% fetal bovine serum (Sigma-Aldrich, 12003C), 0.1 mM dexamethasone (Sigma-Aldrich, D4902), 10 mM β-glycerol phosphate (Sigma-Aldrich, 50020), and 50 mM ascorbate-2-phosphate (Sigma-Aldrich, 49752). After 3 weeks of differentiation, the mineralization capacity of the cells was evaluated by Alizarin Red (Sigma-Aldrich, A5533) staining. For tube formation, we plated Matrigel (BD Biosciences, 354234) in 96-well culture plates and incubated at 37 °C to polymerize for 45 min. We then seeded sorted cells ($2 \times 10^4$ cells/well) on polymerized Matrigel in plates. We cultured the cells with αMEM supplemented with 10% fetal bovine serum. After incubation at 37 °C for 4 h, we observed tube formation by microscopy.

**Statistics.** All statistical analyses were carried out using SPSS 15 software. The data are presented as the mean ± s.d. We performed comparisons using paired, two-tailed $t$-test (for comparison of the concentration of active TGF-β1 or PDGF-BB in osteogenesis and maturation stage in human patients), unpaired, two-tailed Student's $t$-test (for comparison of morphometric analysis of osteogenesis stage and maturation stage of human patients, WT and CED mice, WT and $Csf1^{-/-}$ mice, $LysM$-$cre::Tgfb1^{flox/flox}$ and $Tgfb1^{flox/flox}$ mice, CD68 and pSmad2/3 immunostaining of WT mice four days after Sham and ATP operation, vehicle-treated and antibody-treated CED mice, $caALK2$ mice, vehicle-treated and tamoxifen-treated $Cre$ and $Tgfbr2^{flox/flox}$ mice, and $Tgfbr2^{flox/flox}$ and $Tgfbr2^{-/-}$ mice) or one-way analysis of variance (ANOVA), followed by Tukey's post-hoc test (for all the other comparisons) to determine significance between groups. The level of significance was set at $p < 0.05$. All inclusion/exclusion criteria were pre-established, and no samples or animals were excluded from the analysis. No statistical method was used to predetermine the sample size. The experiments were randomized. The investigators were not blinded to allocation during experiments and outcome assessment.

**Data availability.** The data that support the findings of this study are available within the article and Supplementary Files or available from the corresponding author upon reasonable request.

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

## Acknowledgements

This work was supported in part by the NIH/NIAMS grants AR071432 and AR063943 (to Xu Cao).

## Author contributions

X.W. designed and conducted the majority of the experiments and prepared the manuscript. F.L. conducted some of the surgery, performed CT analyses and helped with manuscript preparation. L.X. helped with flow cytometry analysis. J.C. and G.Z. helped writing the manuscript. Y.M. helped with the *caALK2* mice preparation and analysis. R.D., B.G., H.C. and S.L. helped with *LysM-cre* mice preparation and analysis. P.Y., M.G., M.T. and Y.W. helped with gait analysis. M.W. and C.F. provided suggestions for the project. X.C. developed the concept, supervised the project and wrote most of the manuscript.

## Additional information

**Competing interests:** The authors declare no competing financial interests.

