## [Peer Review File · Nature Communications]

Editorial Note: This manuscript has been previously reviewed at another journal that is not operating a transparent peer review scheme. This document only contains reviewer comments and rebuttal letters for versions considered at Nature Communications. Mentions of prior referee reports have been redacted.

Reviewers' comments:

Reviewer #1 (Remarks to the Author):

The TGF β -b and BMP-signaling related mouse models used in this study are appropriate and valuable genetic models to study the effects of these signaling pathways *in vivo*, and the findings in these models strongly support the author's conclusions.

1D11 is an established pan-TGF β neutralizing antibody previously used for TGF β -inhibition in mice; treatment of control mice with 13C4 (a IgG antibody of the same isotype that does not inhibit TGF β ligands) is an excellent control treatment and has been used by other groups before. It should be added in methods from which company 1D11 and 13C4 were purchased or received. The doses of 1D11 used *in vivo* are in the range used to inhibit TGF β signaling in previous studies in mice. The authors use 3 different doses of 1D11: 5mg/kg, 3x/week IP (Achilles tendon puncture (ATP) and BMP-2/Gelatin Implantation (BGI) model), 5mg/kg, daily IP (CED mice), and 10mg/kg, 3x/week IP (ALK2 mice). It seems that higher doses of 1D11 might be required for a treatment effect in the genetic models, which would not be surprising. If this is indeed the case, a comment on this should be added to the manuscript to calibrate the translatability of the findings.

Other comments:

Line 105 & figure 1i, line 133 & figure 2i : the Elisa kits used to measure TGF β in serum (human R&D DB100B, mouse R&D MB100B) are specific for TGF β 1, but not TGF β 2 and 3. It should be indicated in the manuscript text and figures that the results are for TGF β 1, and that potentially the roles of TGF β 2 and TGF β 3 in heterotopic ossification may be different.

Line 149: "Taken together, the ATP-induced HO mouse model manifests a similar mechanism as observed in the human HO specimens, implying active TGF- β is the driving force for the pathogenesis of HO." – until this point the authors have not presented data suggesting that TGF β is a driver of HO.

Line 303-304: "In HO, effectively mitigates HO progression at broad window during HO progression in different animal models." – this sentence is unclear, please modify

Line 393: Please add in methods from which company 1D11 and 13C4 were purchased or received.

Line 410: may specify "2.3-kb type I collagen promoter" to "2.3-kb Col1a1 promoter"

Reviewer #2 (Remarks to the Author):

The manuscript entitled "Inhibition of Excessive Active TGF- β Attenuates Progression of Heterotopic Ossification" is a well performed study by experts in the field, which adds to our understanding of the molecular and cellular mechanisms that control heterotopic ossification. The

strengths of the paper include the novel finding that transgenic expression of active TGF- β in tendon induces spontaneous HO, as well as rigorous gain and loss of TGF- β function in sophisticated mouse models. As such, the results support the conclusions, and the paper is viewed to be of general interest.

Reviewer #3 (Remarks to the Author):

The authors have adequately addressed prior criticisms regarding TGF β initiation of HO and 1D11 impact on experimental mice mobility (answers #3 and 4 in rebuttal letter and relevant textual changes in the manuscript). However, they should comment in the Discussion about the limitations of current mouse models of FOP and the lack of experimental evidence for the outcome of long-term 1D11 treatment (answers #2 and 1 in rebuttal letter, respectively).

Reviewer #4 (Remarks to the Author):

This is a very interesting and carefully done series of experiments that demonstrate important functional roles for TGF β signaling in endochondral ossification using models of heterotopic ossification. However, the authors seem to imply that TGF β signaling during formation of heterotopic bone is distinct from the normal process of endochondral ossification.

The authors' data provide clear and convincing evidence that TGF β signaling is a key pathway regulating HO in their models, as shown by inhibition with TGF β antibodies, however seem to suggest that TGF β is the critical regulatory event and have not discussed the upstream signals that might lead to activation of TGF β signaling.

Additional specific comments:

1. The authors' explanation (to Reviewer 2, point #2) for HO formation limited to tendons in their CED model should be included in the text. The explanation higher levels of type 1 collagen expression in tendons is the reason for this specificity is not entirely convincing (without additional data to support the statement), however should be included within the text.

2. The authors state that FOP patients do not spontaneously induce HO (line 326). This is an incorrect statement. The authors also state that trauma/venom are required to initiate HO (line 327); this is true in some models, but is not a universally true statement. While one could argue that no HO is 'spontaneous' since in FOP the requirement for an ACVR1 mutation is established, spontaneous HO in FOP is used to refer to episodes of HO with no apparent injury or other clear triggering event. In a mouse model of FOP with the Acvr1 R206H mutation, expression of Acvr1-R206H in Prrx1+ cells results in reproducible post-natal 'spontaneous' HO formation.

3. At several points within the manuscript, the authors seem to overstate conclusions from their data or do not provide the rationale for conclusions. Examples:

- line 111: The authors report increased numbers of bone marrow MSCs in HO bone marrow, then conclude that these cells participate in HO formation without showing evidence for this.

- line 139: What is the evidence to support that CD31-high, Emcn-high vessels are osteogenic? Is there evidence that these vessels specifically promote osteogenic differentiation or only that they are present in tissues undergoing osteogenesis? References should be cited.

- line 1449: That TGF β is a "driving force" for HO pathogenesis is greatly overstated based on data shown.

- lines 162-163: The authors conclude that TGF β secreted by macrophage triggers HO. However, LysM-cre is expressed in the macrophage/monocyte lineage (and therefore is also expressed in the osteoclast lineage) and has been reported to be expressed in neutrophils (see Abram et al., 2014 J

Immunol Methods).

In addition to macrophages, what other immune cells have the authors examined with regard to changes in response to HO induction?

- line 202: The authors conclude that inhibiting TGF β activity at inflammation, chondrogenesis, or osteogenesis stages inhibits HO, however this statement should be qualified to specifically refer to the ATP model since this does not appear to be the case in the BGI model.

- line 320: No data support that HO formation in POH does not involve TGF β signaling.

- lines 319-320: The statement that TGF β signaling inhibition provides a potential treatment for HO in the stage when osteogenesis already occurs, seem to be supported by data in some HO models but not others.

4. The authors state that 90% of Nestin⁺ cells formed cartilage 3 weeks after ATP (line 262). How were these cells quantified and how was this conclusion reached? The authors also state that 70-86% of Nestin⁺ cells participate in type H vessel formation (line 263); how does the same cell attain two cell fates?

Dear editor,

We would like to thank the reviewers for their thoughtful and constructive comments regarding our manuscript. We have addressed all of the questions and comments brought forth through additional clarification and highlighted all changes in the manuscript text file.

Reviewers' comments:

Reviewer #1 (Remarks to the Author):

The TGF β and BMP-signaling related mouse models used in this study are appropriate and valuable genetic models to study the effects of these signaling pathways in vivo, and the findings in these models strongly support the author's conclusions.

1D11 is an established pan-TGF β neutralizing antibody previously used for TGF β -inhibition in mice; treatment of control mice with 13C4 (a IgG antibody of the same isotype that does not inhibit TGF β ligands) is an excellent control treatment and has been used by other groups before. It should be added in methods from which company 1D11 and 13C4 were purchased or received. The doses of 1D11 used in vivo are in the range used to inhibit TGF β signaling in previous studies in mice. The authors use 3 different doses of 1D11: 5mg/kg, 3x/week IP (Achilles tendon puncture (ATP) and BMP-2/Gelatin Implantation (BGI) model), 5mg/kg, daily IP (CED mice), and 10mg/kg, 3x/week IP (ALK2 mice). It seems that higher doses of 1D11 might be required for a treatment effect in the genetic models, which would not be surprising. If this is indeed the case, a comment on this should be added to the manuscript to calibrate the translatability of the findings.

Response: We appreciate the reviewer's encouraging comments. In the revised manuscript, we have added a comment on this.

Other comments:

Line 105 & figure 1i, line 133 & figure 2i: the Elisa kits used to measure TGF β in serum (human R&D DB100B, mouse R&D MB100B) are specific for TGF β 1, but not TGF β 2 and 3. It should be indicated in the manuscript text and figures that the results are for TGF β 1, and that potentially the roles of TGF β 2 and TGF β 3 in heterotopic ossification may be different.

Response: Thanks for the suggestion. In the revised manuscript, we have changed to "TGF- β " to "TGF- β 1".

Line 149: "Taken together, the ATP-induced HO mouse model manifests a similar mechanism as observed in the human HO specimens, implying active TGF- β is the driving force for the pathogenesis of HO." – until this point the authors have not presented data suggesting that TGF β is a driver of HO.

Response: In the revised manuscript, we have changed to "Taken together, the ATP-induced HO mouse model manifests a similar mechanism as observed in the human HO specimens, implying that high concentrations of active TGF β may contribute to the pathogenesis of HO."

Line 303-304: "In HO, effectively mitigates HO progression at broad window during HO progression in different animal models." – this sentence is unclear, please modify

Response: In the revised manuscript, we have changed to “In HO, blockage of active TGF- β effectively mitigates HO progression at broad window during HO progression in different animal models.”

Line 393: Please add in methods from which company 1D11 and 13C4 were purchased or received.

Response: In the revised manuscript, we have added these information in the Methods.

Line 410: may specify “2.3-kb type I collagen promoter” to “2.3-kb Col1a1 promoter”

Response: Thanks for the suggestion. In the revised manuscript, we have changed “2.3-kb type I collagen promoter” to “2.3-kb Col1a1 promoter”.

Reviewer #2 (Remarks to the Author):

The manuscript entitled “Inhibition of Excessive Active TGF- β Attenuates Progression of Heterotopic Ossification” is a well performed study by experts in the field, which adds to our understanding of the molecular and cellular mechanisms that control heterotopic ossification. The strengths of the paper include the novel finding that transgenic expression of active TGF- β in tendon induces spontaneous HO, as well as rigorous gain and loss of TGF- β function in sophisticated mouse models. As such, the results support the conclusions, and the paper is viewed to be of general interest.

Response: We appreciate the reviewer's encouraging comments.

Reviewer #3 (Remarks to the Author):

The authors have adequately addressed prior criticisms regarding TGF β initiation of HO and 1D11 impact on experimental mice mobility (answers #3 and 4 in rebuttal letter and relevant textual changes in the manuscript). However, they should comment in the Discussion about the limitations of current mouse models of FOP and the lack of experimental evidence for the outcome of long-term 1D11 treatment (answers #2 and 1 in rebuttal letter, respectively).

Response: We appreciate the reviewer's encouraging comments. In the revised manuscript, we have added and highlighted the comments in the Discussion.

Reviewer #4 (Remarks to the Author):

This is a very interesting and carefully done series of experiments that demonstrate important functional roles for TGF β signaling in endochondral ossification using models of heterotopic ossification. However, the authors seem to imply that TGF β signaling during formation of heterotopic bone is distinct from the normal process of endochondral ossification.

The authors' data provide clear and convincing evidence that TGF β signaling is a key pathway regulating HO in their models, as shown by inhibition with TGF β antibodies, however seem to suggest that TGF β is the critical regulatory event and have not discussed the upstream signals that might lead to activation of TGF β signaling.

Response: We appreciate the reviewer's encouraging comments. At the early stage of HO, degradation of extra cellular matrix (ECM) or latency associated peptide (LAP) by inflammation may contribute to TGF- β activation¹. During the osteogenesis stage, TGF- β is activated by osteoclastic bone resorption². In the revised manuscript, we have added these in the discussion.

Additional specific comments:

1. The authors' explanation (to Reviewer 2, point #2) for HO formation limited to tendons in their CED model should be included in the text. The explanation higher levels of type 1 collagen expression in tendons is the reason for this specificity is not entirely convincing (without additional data to support the statement), however should be included within the text.

Response: We appreciate the suggestion. In the revised manuscript, we have added the explanation in the text.

2. The authors state that FOP patients do not spontaneously induce HO (line 326). This is an incorrect statement. The authors also state that trauma/venom are required to initiate HO (line 327); this is true in some models, but is not a universally true statement. While one could argue that no HO is 'spontaneous' since in FOP the requirement for an ACVR1 mutation is established, spontaneous HO in FOP is used to refer to episodes of HO with no apparent injury or other clear triggering event. In a mouse model of FOP with the Acvr1 R206H mutation, expression of Acvr1-R206H in Prrx1+ cells results in reproducible post-natal 'spontaneous' HO formation.

Response: Thanks for the correction. In the revised manuscript, we have changed the sentence to “FOP patients with constitutive BMP signaling can induce HO spontaneously, but vigorous ossification occurs with additional stimuli such as inflammation or injuries^{3,4}.”

3. At several points within the manuscript, the authors seem to overstate conclusions from their data or do not provide the rationale for conclusions. Examples:

- line 111: The authors report increased numbers of bone marrow MSCs in HO bone marrow, then conclude that these cells participate in HO formation without showing evidence for this.

Response: In the revised manuscript, we have changed “MSCs participation” to “MSCs numbers”.

- line 139: What is the evidence to support that CD31-high, Emcn-high vessels are osteogenic? Is there evidence that these vessels specifically promote osteogenic differentiation or only that they are present in tissues undergoing osteogenesis? References should be cited.

Response: CD31^{high}Emcn^{high} blood vessels (Type H vessels), which are a specific vessel subtype in bone, have been repeatedly reported to be coupled with osteogenesis specifically⁵⁻¹¹. We cited these references in line 86-89.

- line 149: That TGFβ is a “driving force” for HO pathogenesis is greatly overstated based on data shown.

Response: In the revised manuscript, we have changed to “High concentrations of active TGF- β may contribute to the pathogenesis of HO.”

- lines 162-163: The authors conclude that TGF β secreted by macrophage triggers HO. However, *LysM-cre* is expressed in the macrophage/monocyte lineage (and therefore is also expressed in the osteoclast lineage) and has been reported to be expressed in neutrophils (see Abram et al., 2014 J Immunol Methods). In addition to macrophages, what other immune cells have the authors examined with regard to changes in response to HO induction?

Response: In the revised manuscript, we have made the following changes and highlighted in the text.

- 1) Line 155, change “Elevated TGF- β secreted by macrophage triggers HO” to “Elevated TGF- β at the inflammation stage triggers HO”.
- 2) Line 156-158, change “We found that accumulation of macrophage and high levels of active TGF- β two days after HO induction in ATP mice (Fig. 3a-d), suggesting that active TGF- β and macrophages are closely related to the onset of HO.” to “We found that accumulation of CD68⁺ immune cells and high levels of active TGF- β two days after HO induction in ATP mice (Fig. 3a-d), suggesting that active TGF- β and immune cells are closely related to the onset of HO.”
- 3) Line 163-165, change ““We then generated a *LysM-cre::Tgfb1^{lox/lox}* mouse model (*Tgfb1^{-/-}*) where macrophage ” to “We then generated a *LysM-cre::Tgfb1^{lox/lox}* mouse model (*Tgfb1^{-/-}*) where immune cells, including macrophage/monocyte lineage and neutrophils^{12,13}”
- 4) Line 167-168, change “Therefore, it is the elevated TGF- β secreted by macrophage that

triggers HO.” to “Therefore, it is the elevated TGF- β secreted by immune cells at the inflammation stage that triggers HO.”

5) Apart from examination of CD68⁺ immune cells, macrophage deficiency *Csf1*^{-/-} mice and *LysM-cre::Tgfb1*^{flx/flx} mice where immune cells, including macrophage/monocyte lineage and neutrophils no longer produce TGF β 1, we did not test other immune cells.

- line 202: The authors conclude that inhibiting TGF β activity at inflammation, chondrogenesis, or osteogenesis stages inhibits HO, however this statement should be qualified to specifically refer to the ATP model since this does not appear to be the case in the BGI model.

Response: In BGI model, heterotopic bone was formed by 2 weeks after implantation and enlarged by 4 weeks (Supplementary Fig. 4a, b). H&E and SOFG staining showed calcified cartilage formed at 2 weeks post-implantation and a fully developed ectopic bone with marrow at 4 weeks (Supplementary Fig. 4c, d) suggesting a mixed stage of chondrogenesis and osteogenesis at week 2 and maturation stage at week 4. Therefore, blockage of TGF- β activity from week 2 inhibited active TGF- β from both chondrogenesis and osteogenesis stages. This inhibition successfully attenuated HO formation. Therefore, our conclusion of HO mitigation by inhibiting TGF- β activity at inflammation, chondrogenesis, or osteogenesis stages is qualified to BGI model.

- line 320: No data support that HO formation in POH does not involve TGF β signaling.

Response: The objective of present study is to understand the pathomechanism of acquired heterotopic endochondral ossification (HO). We found high levels of active TGF- β initiates and drives the progression of HO. This study also provides a potential treatment for HO. POH is a process of intramembranous bone formation by null mutations in *GNAS* through activation of Hedgehog signaling¹⁴, a distinct procedure from endochondral ossification. We agree with the reviewer's comments, as we do not have any supporting data, POH may also involve TGF β signaling. Further mechanistic studies are needed, but beyond the scope of the current study. We softened our tune by changing "does not seem involving TGF- β signaling." to "may not involve TGF- β signaling" in line 336.

- lines 319-320: The statement that TGF β signaling inhibition provides a potential treatment for HO in the stage when osteogenesis already occurs, seem to be supported by data in some HO models but not others.

Response: We have responded this question. Please see question and response of "- line 202".

4. The authors state that 90% of Nestin⁺ cells formed cartilage 3 weeks after ATP (line 262). How were these cells quantified and how was this conclusion reached? The authors also state that 70-86% of Nestin⁺ cells participate in type H vessel formation (line 263); how does the same cell attain two cell fates?

Response: As we have answered in the responses to Reviewer 2, point #3, Nestin⁺ cells are heterozygous populations providing the precursors to mesenchymal and endothelial lineages¹⁵⁻¹⁸. The sub-population of mesenchymal precursors could contribute to chondrogenesis and endothelial precursors to vessel formation during HO development. At

different microenvironment, Nestin⁺ cells performed differently¹⁹. At the chondrogenesis stage, more than 90% Nestin lineage cells formed cartilage 3 weeks (Fig. 6 f, g), while 70-86% of Nestin lineage cells participated in type H vessel at the osteogenesis stage (Fig. 6h, i). For quantification of Nestin lineage cells, we used *Nestin-creERT2::R26R-EYFP* mice to trace Nestin lineage cells and all YFP⁺ cells were considered as Nestin lineage cells. For chondrocytes quantification, we calculated all the cells in red (COLII⁺) area and considered them as COLII⁺ chondrocytes. The YFP⁺COLII⁺ cells were considered as chondrocytes that derived from Nestin⁺ cells. We found more than 90% of YFP⁺ cells were YFP⁺COLII⁺ cells, therefore, we drew the conclusion that more than 90% Nestin lineage cells formed cartilage (Fig. 6 f, g).

1. Annes, J.P., Munger, J.S. & Rifkin, D.B. Making sense of latent TGFbeta activation. *J Cell Sci* **116**, 217-224 (2003).
2. Tang, Y., *et al.* TGF-beta1-induced migration of bone mesenchymal stem cells couples bone resorption with formation. *Nature medicine* **15**, 757-765 (2009).
3. Shore, E.M. & Kaplan, F.S. Inherited human diseases of heterotopic bone formation. *Nature reviews. Rheumatology* **6**, 518-527 (2010).
4. Kaplan, F.S., Pignolo, R.J. & Shore, E.M. Granting immunity to FOP and catching heterotopic ossification in the Act. *Semin Cell Dev Biol* **49**, 30-36 (2016).
5. Gerhardt, H. & Betsholtz, C. Endothelial-pericyte interactions in angiogenesis. *Cell Tissue Res* **314**, 15-23 (2003).
6. Gerhardt, H. & Semb, H. Pericytes: gatekeepers in tumour cell metastasis? *J Mol Med (Berl)* **86**, 135-144 (2008).
7. Armulik, A., Abramsson, A. & Betsholtz, C. Endothelial/pericyte interactions. *Circ Res* **97**, 512-523 (2005).
8. Gaengel, K., Genove, G., Armulik, A. & Betsholtz, C. Endothelial-mural cell signaling in vascular development and angiogenesis. *Arterioscler Thromb Vasc Biol* **29**, 630-638 (2009).
9. Kusumbe, A.P., Ramasamy, S.K. & Adams, R.H. Coupling of angiogenesis and osteogenesis by a specific vessel subtype in bone. *Nature* **507**, 323-328 (2014).
10. Xie, H., *et al.* PDGF-BB secreted by preosteoclasts induces angiogenesis during coupling with osteogenesis. *Nature medicine* **20**, 1270-1278 (2014).
11. Lories, R.J. & Luyten, F.P. The bone-cartilage unit in osteoarthritis. *Nature reviews. Rheumatology* **7**, 43-49 (2011).

12. Clausen, B.E., Burkhardt, C., Reith, W., Renkawitz, R. & Forster, I. Conditional gene targeting in macrophages and granulocytes using LysMcre mice. *Transgenic Res* **8**, 265-277 (1999).
13. Abram, C.L., Roberge, G.L., Hu, Y. & Lowell, C.A. Comparative analysis of the efficiency and specificity of myeloid-Cre deleting strains using ROSA-EYFP reporter mice. *J Immunol Methods* **408**, 89-100 (2014).
14. Regard, J.B., *et al.* Activation of Hedgehog signaling by loss of GNAS causes heterotopic ossification. *Nature medicine* **19**, 1505-1512 (2013).
15. Mendez-Ferrer, S., *et al.* Mesenchymal and haematopoietic stem cells form a unique bone marrow niche. *Nature* **466**, 829-834 (2010).
16. Ono, N., *et al.* Vasculature-associated cells expressing nestin in developing bones encompass early cells in the osteoblast and endothelial lineage. *Developmental cell* **29**, 330-339 (2014).
17. Itkin, T., *et al.* Distinct bone marrow blood vessels differentially regulate haematopoiesis. *Nature* **532**, 323-328 (2016).
18. Suzuki, S., Namiki, J., Shibata, S., Mastuzaki, Y. & Okano, H. The neural stem/progenitor cell marker nestin is expressed in proliferative endothelial cells, but not in mature vasculature. *The journal of histochemistry and cytochemistry : official journal of the Histochemistry Society* **58**, 721-730 (2010).
19. Kunisaki, Y., *et al.* Arteriolar niches maintain haematopoietic stem cell quiescence. *Nature* **502**, 637-643 (2013).

Reviewers' comments:

Reviewer #4 (Remarks to the Author):

The authors have been very responsive to previous comments and questions.

This is an interesting and important set of data that will increase understanding the regulation of heterotopic bone formation. The authors are requested to clarify a few further points.

1. Although some of the assays are specific for TGFβ₁, the 1D11 antibody is reported to block TGFβ_{1,2,3}. The authors are requested to provide this information about the 1D11 antibody in the Methods, as well as whether the antibody blocks the receptor through binding to the extracellular or cytoplasmic domain of the receptor. This is relevant to the use of the antibody in the caAlk2 mouse model and data interpretation since this constitutively active receptor is often described as ligand-independent.

2. This reviewer had previously commented:

The authors state that 90% of Nestin+ cells formed cartilage 3 weeks after ATP (line 262). How were these cells quantified and how was this conclusion reached? The authors also state that 70-86% of Nestin+ cells participate in type H vessel formation (line 263); how does the same cell attain two cell fates?

The authors have provided a thoughtful response to this point, however perhaps I did not phrase my specific question clearly enough. I will rephrase:

Do the authors mean to convey that 90% of cartilage cells were derived from the Nestin+ lineage or that 90% of the Nestin+ lineage cells became cartilage? The authors' statements seem to indicate the latter, but neither the images or the graphs provided are sufficient or sufficiently clear to make an independent determination. Clarification of the authors statements and re-labeling of the graphs - along with a more detailed explanation in the Methods or figure legend about how the data were quantified - are requested.

3. Similar clarification is requested for the statements that 90% of Nestin lineage cells formed cartilage at 3 weeks but that 70-86% of Nestin lineage cells become type H vessel cells at later stages. If 90% of the cells initially attained a cartilage cell fate, but later 70-86% of the Nestin+ cells became type H cells, then it appears that the authors are concluding that a large number of the cartilage cells transdifferentiated to type H cells. If this is the authors' intended conclusion, they are requested to state this clearly. If not, then rephrasing is necessary.

Additional points

4. Line 56. Although the cited report used Gnas null mutations in a mouse model to induce heterotopic ossification, the GNAS mutations in POH patients are heterozygous inactivating mutations. (See NEJM 2002, 346: 99.)

5. Line 215. Reference 37 is a meeting abstract; since the data supporting the statement that "inhibition of BMP signaling did not mitigate HO progression after the chondrogenesis stage" cannot be examined, it is suggested that the authors revise the sentence to be less dogmatic.

6. Lines 275-279. The authors are requested to identify the tissue or tissues examined for the Scx lineage cells – tendon, and/or other tissues?

7. Line 293. The data from the experiments described in this section of the manuscript

demonstrate that TGFβ is important, but cannot make conclusions that elevated amounts are required.

8. Line 326. The authors perhaps meant 'heterogeneous' and not 'heterozygous'

9. Line 346. The authors refer to ALK2-Q206H; the mutation in FOP is ALK2-R206H. Also, the ALK2-Q207D mouse model does not fully mimic human FOP not because of "limited understanding of the human condition", but because, mechanistically, it is a strong constitutively activating mutation that induces much higher levels of pSmad1/5/8 signaling compared the the much more mildly activating R206H mutation.

10. Line 356. The phrase "by contrast" seems inappropriate since there is no clear evidence that TGFβ inhibition will be more effective than the other therapeutic strategies described.

We would like to thank the reviewer #4's thoughtful and constructive comments regarding our manuscript. We have addressed all of the questions and comments through additional clarification and highlighted all changes in the manuscript text file.

Reviewer #4's comments:

The authors have been very responsive to previous comments and questions

This is an interesting and important set of data that will increase understanding the regulation of heterotopic bone formation. The authors are requested to clarify a few further points.

1. Although some of the assays are specific for TGF β 1, the 1D11 antibody is reported to block TGF β 1,2,3. The authors are requested to provide this information about the 1D11 antibody in the Methods, as well as whether the antibody blocks the receptor through binding to the extracellular or cytoplasmic domain of the receptor. This is relevant to the use of the antibody in the caAlk2 mouse model and data interpretation since this constitutively active receptor is often described as ligand-independent.

Response: 1D11 is a monoclonal antibody that neutralizes 3 major active TGF- β isoforms (TGF- β 1, -2, and -3), the known ligands for the TGF- β receptor kinase. It does not block the receptor or bind other ligands in the TGF- β superfamily, including activin or BMP¹. We have added this information in Methods in the revised manuscript (line 426 - 429).

The reviewer is correct. 1D11 antibody does not have any effects on signaling of caALK2. However, there is no spontaneous HO in caALK2 mice, and HO occurs only with a combination of trauma and cobra venom or cardiotoxin^{2,3}, indicating other factor(s) essential for HO initiation. The interpretation of 1D11 antibody inhibition of HO in caALK2 mice is that 1D11 antibody neutralizes TGF- β ligands at inflammation stage essential for HO initiation. Accordingly, this interpretation has been added in Discussion in the revised manuscript (line 344-346).

2. This reviewer had previously commented:

The authors state that 90% of Nestin⁺ cells formed cartilage 3 weeks after ATP (line 262). How were these cells quantified and how was this conclusion reached? The authors also state that 70-86% of Nestin⁺ cells participate in type H vessel formation (line 263); how does the same cell attain two cell fates?

The authors have provided a thoughtful response to this point, however perhaps I did not phrase my specific question clearly enough. I will rephrase:

Do the authors mean to convey that 90% of cartilage cells were derived from the Nestin⁺ lineage or that 90% of the Nestin⁺ lineage cells became cartilage? The authors' statements seem to indicate the latter, but neither the images or the graphs provided are sufficient or sufficiently clear to make an independent determination. Clarification of the authors statements and re-labeling of the graphs - along with a more detailed explanation in the Methods or figure legend about how the data were quantified - are requested.

Response: We apologize for the confusing statement for the Nestin lineage tracing experiment. We have rephrased the statement as “90% of cartilage cells were derived from the Nestin⁺ lineage”. We have added quantification method from line 529 to 535.

3. Similar clarification is requested for the statements that 90% of Nestin lineage cells formed cartilage at 3 weeks but that 70-86% of Nestin lineage cells become type H vessel cells at later stages. If 90% of the cells initially attained a cartilage cell fate, but later 70-86% of the Nestin⁺ cells became type H cells, then it appears that the authors are concluding that a large number of the cartilage cells transdifferentiated to type H cells. If this is the authors' intended conclusion, they are requested to state this clearly. If not, then rephrasing is necessary.

Response: Again, we apologize for the confusing statement. We have rephrased the statement as “70-86% of type H vessel were formed by Nestin lineage cells 6 weeks after ATP”.

Additional points:

4. Line 56. Although the cited report used Gnas null mutations in a mouse model to induce heterotopic ossification, the GNAS mutations in POH patients are heterozygous inactivating mutations. (See NEJM 2002, 346: 99.)

Response: In the revised manuscript, we have rephrased to “POH is a process of intramembranous bone formation by null or heterozygous mutations in *GNAS*^{4,5}.”

5. Line 215. Reference 37 is a meeting abstract; since the data supporting the statement that “inhibition of BMP signaling did not mitigate HO progression after the chondrogenesis stage” cannot be examined, it is suggested that the authors revise the sentence to be less dogmatic.

Response: As suggested, we have deleted this sentence.

6. Lines 275-279. The authors are requested to identify the tissue or tissues examined for the Scx lineage cells – tendon, and/or other tissues?

Response: As suggested, we have specified “Scx⁺ cells in tendon”.

7. Line 293. The data from the experiments described in this section of the manuscript demonstrate that TGFβ is important, but cannot make conclusions that elevated amounts are required.

Response: In the revised manuscript, we have rephrased “elevated TGF-β” as “active TGF-β”.

8. Line 326. The authors perhaps meant ‘heterogeneous’ and not ‘heterozygous’

Response: In the revised manuscript, we have corrected this mistake.

9. Line 346. The authors refer to ALK2-Q206H; the mutation in FOP is ALK2-R206H. Also, the ALK2-Q207D mouse model does not fully mimic human FOP not because of “limited understanding of the human condition”, but because, mechanistically, it is a strong constitutively activating mutation that induces much higher levels of pSmad1/5/8 signaling compared the much more mildly activating R206H mutation.

Response: Thanks for the suggestion. In the revised manuscript, we have added this statement in our discussion (line 347 - 349).

10. Line 356. The phrase “by contrast” seems inappropriate since there is no clear evidence that TGF β inhibition will be more effective than the other therapeutic strategies described.

Response: In the revised manuscript, we have rephrased “by contrast” as “In present study, we found...”.

1. Dasch, J.R., Pace, D.R., Waegell, W., Inenaga, D. & Ellingsworth, L. Monoclonal antibodies recognizing transforming growth factor-beta. Bioactivity neutralization and transforming growth factor beta 2 affinity purification. *J Immunol* **142**, 1536-1541 (1989).
2. Fukuda, T., *et al.* Generation of a mouse with conditionally activated signaling through the BMP receptor, ALK2. *Genesis* **44**, 159-167 (2006).
3. Medici, D., *et al.* Conversion of vascular endothelial cells into multipotent stem-like cells. *Nature medicine* **16**, 1400-1406 (2010).
4. Shore, E.M., *et al.* Paternally inherited inactivating mutations of the GNAS1 gene in progressive osseous heteroplasia. *N Engl J Med* **346**, 99-106 (2002).
5. Regard, J.B., *et al.* Activation of Hedgehog signaling by loss of GNAS causes heterotopic ossification. *Nature medicine* **19**, 1505-1512 (2013).

Reviewers' Comments:

Reviewer #4:

Remarks to the Author:

The authors have been quickly and well responsive to previous comments, however two points were not completely addressed.

1. Please note the uncorrected typo on line 347 (previously noted in comment referring to line 346):

The mutation in FOP is ALK2-R206H not ALK2-Q206H.

2. Line 56: Only GNAS heterozygous mutations have been found in POH patients; GNAS null mutations have not been documented in humans. Mouse models of POH have used homozygous Gnas null alleles to induce heterotopic ossification. These points are unclear in the manuscript.

Reviewer #4 (Remarks to the Author):

The authors have been quickly and well responsive to previous comments, however two points were not completely addressed.

1. Please note the uncorrected typo on line 347 (previously noted in comment referring to line 346): The mutation in FOP is ALK2-R206H not ALK2-Q206H.

Response: In the revised manuscript, we have corrected this mistake.

2. Line 56: Only GNAS heterozygous mutations have been found in POH patients; GNAS null mutations have not been documented in humans. Mouse models of POH have used homozygous Gnas null alleles to induce heterotopic ossification. These points are unclear in the manuscript.

Response: In the revised manuscript, we have changed to “POH is a process of intramembranous bone formation by heterozygous mutations in GNAS in patients.”